# Advances and Novel Perspectives on Colloids, Hydrogels, and Aerogels Based on Coordination Bonds with Biological Interest Ligands

**DOI:** 10.3390/nano11071865

**Published:** 2021-07-20

**Authors:** Noelia Maldonado, Pilar Amo-Ochoa

**Affiliations:** 1Department of Inorganic Chemistry, Universidad Autónoma de Madrid, E-28049 Madrid, Spain; 2Institute for Advanced Research in Chemistry (IAdChem), Universidad Autónoma de Madrid, E-28049 Madrid, Spain

**Keywords:** colloids, hydrogels, aerogels, coordination compounds, nucleobases, amino acids

## Abstract

This perspective article shows new advances in the synthesis of colloids, gels, and aerogels generated by combining metal ions and ligands of biological interest, such as nucleobases, nucleotides, peptides, or amino acids, among other derivatives. The characteristic dynamism of coordination bonds between metal center and biocompatible-type ligands, together with molecular recognition capability of these ligands, are crucial to form colloids and gels. These supramolecular structures are generated by forming weak van der Waals bonds such as hydrogen bonds or π–π stacking between the aromatic rings. Most gels are made up of nano-sized fibrillar networks, although their morphologies can be tuned depending on the synthetic conditions. These new materials respond to different stimuli such as pH, stirring, pressure, temperature, the presence of solvents, among others. For these reasons, they can trap and release molecules or metal ions in a controlled way allowing their application in drug delivery as antimicrobial and self-healable materials or sensors. In addition, the correct selection of the metal ion enables to build catalytic or luminescent metal–organic gels. Even recently, the use of these colloids as 3D-dimensional printable inks has been published. The elimination of the solvent trapped in the gels allows the transformation of these into metal–organic aerogels (MOAs) and metal–organic xerogels (MOXs), increasing the number of possible applications by generating new porous materials and composites useful in adsorption, conversion, and energy storage. The examples shown in this work allow us to visualize the current interest in this new type of material and their perspectives in the short-medium term. Furthermore, these investigations show that there is still a lot of work to be done, opening the door to new and interesting applications.

## 1. Introduction

Gel-like soft materials are colloidal systems which comprise two coexisting phases, a solid one that is expanded throughout its volume entirely by a liquid phase, immobilized inside the cross-linked solid phase. These materials can be classified in different ways.

(i)According to the nature of solvents such as hydrogels or organogels.(ii)Depending upon the force driving their molecular aggregation. In this sense, the gel is classified as a physical or supramolecular gel when gelation is caused by intermolecular non-covalent interactions, such as hydrogen bonding, hydrophobic interactions, dipolar interactions, electrostatic interactions, and π–π stacking. In contrast, when covalent bonds drive the cross-linking, the gel is defined as a chemical gel.(iii)There are also gels depending on their composition, such as purely inorganic, which are mainly made up of metal nanoparticles and metal oxides. Inorganic and organic hybrids, where organic molecules are introduced to the aforementioned oxides. Purely organic, and finally, around 2004, new coordination compound gels and aerogels began to appear called metal–organic gels (MOGs), or coordination polymer gels (CPGs) [1].

MOGs and CPGs have expanded new approaches horizons beyond the synthesis of new materials by changing the building blocks. These new soft materials with identical chemical composition as the bulk material but drastically different physical properties can be generated by playing with the synthetic conditions.

Hydrogels have to overcome some barriers yet to be successful overall related to the tailoring of their chemical properties, which are important to generate functional materials. In most cases, hydrogel matrices are attractive in biological applications such as drug delivery and wound healing due to their cytocompatibility, high porosity, and water content. However, in other practical applications such as catalysis and adsorption, the architecture of materials has to be free of liquid in the macroscale. This can be achieved through specific drying methods such as supercritical drying and freeze-drying, which lead to the formation of the so-called aerogels, whose structural network is filled with gas. In addition, the world of gels and aerogels is intimately related to nanomaterials science since their obtaining mainly implies bottom-up approaches, which allow the creation of fibers, ribbons, and cylinders nanostructures, among others. They have deeply been studied for more than 80 years with an extended materials list continuously growing, looking forward to finding the adequate synthetic conditions from any chemical compound obtaining their gel and the respective aerogel.

There are outstanding revision works that explain in more detail what these types of materials are [2,3,4,5], what are the standard or newest techniques to obtain them, and what are their principal applications [6,7,8,9]. This work aims to show the recent advances obtained by combining metal ions with biologically relevant ligands such as nucleobases [10], nucleotides, oligonucleotides, amino acids, and peptides to mitigate the shortcoming of individual components. An advantage of this type of metal–organic gels is its possibility of spontaneous gelation without ultraviolet (UV) light or chemical initiators, thanks to the molecular recognition capabilities of these molecules. For this reason, the reactions can be made under mild and physiological conditions. However, these synthesis conditions prevent obtaining quality single crystals, so the knowledge of real solid-state structures is a challenge in most metal–organic gels/aerogels, which end up being proposed [11,12,13]. This kind of new bioinspired coordination compounds will lead to the development of biomimetic and biocompatible metal–organic gels/aerogels [14,15] with low toxicity. Furthermore, this new type of metal–organic gels or aerogels will likely provide innovative applications in 3D printing technologies [16,17] or as nanocarrier systems [18].

## 2. Colloid Dispersions Based on Coordination Bonds with Interesting Biological Ligands

Within chemistry, the world of colloids is quite broad and attracts great interest from medical, biological, food, or environmental perspectives [19,20,21]. They play a crucial role in human health, nutrition, and the environment since they can transport nutrients and transfer pollutants. The dispersed nanoparticles are suspended throughout another phase without being settled to the bottom of the container for long periods. A colloid must have particle dimensions between 1–1000 nanometers, which means the dispersed particles in the phase have to be larger than a molecule’s size but smaller than what can be seen with a naked eye. If the dimensions are smaller than this, the substance is considered a solution, and if they are larger than the substance, it is a suspension. In the following examples that we are going to mention here, the nanofibers or nanoparticles dispersed in water are coordination compounds/polymers with ligands of biological interest stabilized by supramolecular interactions, generally Van der Waals forces, and which by means of stimuli can be transformed into the corresponding hydrogels by the sol−gel transition.

### 2.1. Nucleobases and Nucleotides for the Creation of Colloids Useful as Luminescent, Nanocarriers, Or 3D Printed Sensors

Nucleobases, nucleosides, and nucleotides are extremely interesting ligands with extraordinary molecular recognition capability to form complex supramolecular structures. They have a strong capacity to form π–π stacking and hydrogen bonds. Also, they have extraordinary metal-ion-binding abilities and can easily modify for this purpose. Therefore, they are helpful in colloids and gel formation (Figure 1) [22].

In these novel materials, incorporation of metal ions is an effective method for establishing extra-interactions among the building blocks and consequently trigger colloid or gel formation, being considered a straightforward way to integrate the specific properties of metals (conductivity, color, emission, magnetism, among others) with the colloid/gel/aerogel properties [23]. The use of metal–organic gels has been little studied, probably due to the high focus on gels and aerogels derived from organic polymers and purely inorganic materials, due to the initial difficulty of finding and understanding their main formation mechanisms.

Recent works have shown that colloidal suspensions of nanoparticles, nanofibers, or nanoribbons can be transformed into the corresponding gels through incubation times or the application of physical stimuli (shaking, temperature). As a couple of examples, J. Dash et al. exploited the chelating ability of Ag(I) ions with guanosine monophosphate (GMP) to produce a series of supramolecular hydrogels based on the spontaneous self-assembly of Ag-GMP nanofilaments in water [24]. The obtained hydrogels consist of helically stacked nanofilaments with a gel-to-sol transition between 50–80 °C, as differential scanning calorimetry (DSC) measurements showed depending on the matrix composition. Rheological properties are strongly linked to the influence of the molar ratio between metal centers and GMP ligands (see Section 3.1). Thanks to this, protein molecules like cytochrome c could be immobilized without loss of their function, and cationic dyes such as Hoechst-33258 induced chirality transfer and disrupted the hydrogel superstructure.

Li Xu et al. [25] have already reported that lanthanides-adenosine monophosphate (AMP) at 2:1 ratio formed two models of coordination materials: the lightest lanthanides from La(III) to Tb(III) formed nanoparticles, while the heaviest ones initially formed nanoparticles followed by its spontaneously transformation to hydrogels (Figure 2). In the beginning, colloidal suspensions of nanoparticles were obtained, and overnight, from Dy(III) to Lu(III), all became hydrogels. The gels were formed by one unique sol−gel transition, where the morphology of the initially produced nanoparticles turned into nanofibers. These transformations depend on the critical size of lanthanide ions and, because of this, gelation in the heaviest lanthanides was favored, and the duration of the process was different. In addition, monophosphate and lanthanide species were critical as building blocks of the gels, which were stable under slightly acidic pH (see Section 3.1).

More recently, perspectives on the use of metal–organic colloids are changing, as it has been demonstrated by Vegas et al. [18], who showed the synthesis of a 1D-Cu(II) antiferromagnetic coordination polymer (CP) with modified thymine (thymine-1-acetic acid, TAcOH). Thanks to the insolubility in the aqueous medium where it is obtained, this 1D-CP can be generated in nanoribbons forming a stable colloidal suspension for long periods. This colloid is made up of nanoribbons a few microns long, 100–150 nm wide and heights of 40–60 nm (Figure 3C). Its stability at physiological pHs means that this colloid can be incubated with different cell lines to verify the low cytotoxicity of the CP obtained, even considering the presence of Cu(II) ions. The positive results in the viability study (Figure 3B) allowed to incubate this compound with synthetic oligonucleotides chains. The results showed that the thymine residues present along the CP chain (Figure 3A) had selectivity for adenine oligonucleotide (Figure 3D). This result opened the door to the possible use of this CP as a cellular nanocarrier.

The main role of this colloidal coordination polymer did not remain in the previous work. Its colloid stability, nanometric size, solvato- and thermochromic behavior against temperature, and some organic solvents allowed the creation of 3D printing composite material with moisture sensing capabilities (Figure 4) [17]. This embedded CP has a remarkable water sensitivity in air and organic solvents (methanol, tetrahydrofuran, ethanol, and acetonitrile) (Figure 4A). Its sensing capacity is connected to its structural transformation due to the water molecules loss with temperature (60 °C) or by solvent molecules competition, which induces a relevant color change from blue to violet, whilst the color returns from violet to blue with the ambient moisture (Figure 4B). This process was carefully analyzed through theoretical calculations and advanced X-ray techniques. Both the powder and 3D printed objects are stable on-air over 12 months at biological pHs, suggesting possible applications as robust colorimetric sensors. These results prepare the way to create a family of new 3D printed materials based on the incorporation of multifunctional coordination polymers in stable colloids with organic polymers.

## 3. Hydrogels Based on Coordination Bonds with Bioinspired Ligands

Gels are very useful forms because, given their physical and rheological properties, they are practical to use overall in the pharmaceutical industry. According to the solvent’s nature, gels can be classified as hydrogel or organogel if the solvent is water or organic solvent(s), respectively. As already indicated, hydrogels are self-assembled in water to form 3D supramolecular networks, encapsulating abundant water. Regarding the gelation driving forces, gels can be grouped into physical gels where the intermolecular interactions are responsible for gel formation and chemical gels, where the gel skeleton is cross-linked by covalent and/or coordination bonds [26]. Considering water is the only solvent necessary to sustain life on Earth, it is not surprising that hydrogel applications in life sciences have made significant progress [27,28,29,30,31]. In the following lines, we are going to expose the advances in the MOGs related to amino acids, peptides, nucleobases, and mainly transition metal ions such as Zn(II), Cu(II), Ni(II) or Co(II) [32].

### 3.1. Oligonucleotides, Nucleotides, and Nucleobases as Gelators for the Creation of Novel Stimuli–Response and Self-Healing Metal–Organic Hydrogels for Drug Delivery, Wastewater Treatment, Catalysis and Antibacterial Applications

The use of adenosine, guanosine, or thymine and uracil derivatives as interesting gelators is mainly due to their pH tautomeric capacity and molecular recognition abilities [33,34]. J. Dash et al. published some attractive Ag(I) GMP hydrogels in 2011, with different structure and viscoelastic properties depending on the GMP:Ag molar ratio (Figure 5) [24]. 1:1.5 molar ratio gels showed that the storage moduli presented an elastic response and larger values than the loss moduli, confirming the presence of an insistent solid-like viscoelastic hydrogel network. Gels with 1:1.2 ratio had the highest storage modulus value. A similar tendency in both moduli was observed to the 1:1 and 1:1.25 ratios but at lower frequencies and lower stress, <0.01–1 Hz and 20 Pa instead of 102 Hz and 100 Pa for 1:1.5 molar ratio. These last gels also presented a viscous fluid transformation with shear stress above 10 Pa. Hydrogel formation was related to a pH reduction, consistent with Ag(I) ions binding to the guanosine moiety and the protons release due to enol formation. On the other hand, photoreduction of Ag(I) ions is associated with the nanofilaments, resulting in the construction of nanocomposite hydrogels impregnated with plasmonic Ag nanoparticles by prolonged exposure to light. Finally, the molecular recognition ability of the Ag-GMP nanofilaments was also investigated using water-soluble dyes such as methylene blue or fluorescein that were easily incorporated into the Ag-GMP hydrogels to generate materials with optical and fluorescent properties. The authors suggested possible uses of these soft materials in controlled drug release and other molecular recognition-based applications.

In 2015, Hao Liang et al. obtained a supramolecular gel formed by coordination of Zn(II) with adenosine monophosphate (AMP) (Figure 6). In this case, the gelation was induced by mechanical forces [35]. This hydrogel was selectively formed with AMP, in contrast with GMP and CMP. The Zn(II) coordination with adenine seemed crucial for forming the gel complex, and their ultraviolet–visible (UV–vis) data indicated the Zn-AMP binding is the strongest of the three. Although the type of nucleobase was crucial, phosphates number in the molecule seemed to have influence, being essential to have just one phosphate linked to the nucleotide. A stimulating observation was that the mechanically disrupted gels could be easily re-formed. Indeed, with mechanical agitation by a vortex mixer, the Zn-AMP gel was converted into small gel pieces (like a sol phase), yielding a turbid suspension which was put back together after sonication. This mechanically induced gel transition can be repeated several times, and this property might help entrap guest molecules in the gels and remove them from water.

As we have already mentioned in the colloid section (see Section 2.1), six new robust lanthanides adenosine monophosphate (AMP) amorphous hydrogels have been obtained after overnight incubation from the nanoparticles colloidal suspensions, with pH stimulus-response according to the AMP protonation/deprotonation [25]. These hydrogels (Figure 7) are spontaneously formed with the heaviest of the group (Dy(III) to Lu(III)) after initial nanoparticles formation, while the lightest lanthanides (La(III) to Tb(III)) only form nanoparticles. This slow sol-to-gel transition is accompanied by heat release, as indicated by isothermal titration calorimetry. In the reticulation process, the researchers proposed that heavy lanthanides bind preferentially to the phosphate groups forming a structure more kinetically stable than thermodynamically. Applying thermal energy, the system became thermodynamically more stable by replacing some of the phosphate bindings with base binding, freeing part of phosphate groups. Surprisingly, this did not happen with GMP, and again gels were not formed as in the case previously reported by Hao Liang et al. [35]. The contribution of phosphate binding to the lanthanides can be more significant than those of AMP, being reflected in the positive enthalpy of heavy lanthanides. These novel gels were used to successfully encapsulate the glucose oxidase enzyme (GOx) with better stability than the Ln-AMP nanoparticles.

Anil Kumar et al., have used akaganeite (β-FeOOH) as a possible iron oxide nanostructure to integrate it together with AMP in order to combine the superparamagnetic properties of this oxide, its tunnel-like structure, and low toxicity, together with the ability of the nucleotide to generate supramolecular interactions and form a colloid that undergoes, with an increase in AMP concentration, into a spontaneous and reversible (by stirring) transformation (around 10 days) to the corresponding porous and superparamagnetic hydrogel. In this work, they confirmed that gelation is triggered by phosphate groups and sugar moiety, so that the AMP interaction with the β-FeOOH core can be manipulated by a change in reaction parameters. The modification of the AMP-β-FeOOH ratio allows aggregations of NPs and nanoribbons to be obtained with different sizes, and morphologies, found Fe (III) and Fe_3_O_4_ as a function of the reaction conditions. The results obtained from the investigation show that the gel is formed by a β-FeOOH core, coated with AMP. In addition, the porosity of the hydrogel can be 2.5 times greater than that of β-FeOOH alone [36].

In 2016, Ag(I) ions were combined with unsubstituted nucleobases to form metal–organic pH-responsive hydrogels [37]. In this study, nanofibrillar metal–organic gels are obtained under deprotonated conditions with adenine (Ag-A), cytosine (Ag-C), thymine (Ag-T), and uracil (Ag-U). The gelation process between guanine and Ag(I) occurred only under acidic conditions (Ag-pG) via the self-organization of cubic metal–organic particles (Figure 8). As usual in this sort of material, obtaining single crystals of Ag-nucleobases metal–organic gels for understanding its exact coordination environment were not successful. Therefore, density functional theory (DFT) studies were carried out to propose a solid-state structure (Figure 8e). As silver and its compounds are known to exhibit a broad bactericidal spectrum, these Ag-nucleobase hydrogels were tested against the Gram-negative, rod-shaped bacteria *Escherichia coli* and the Gram-positive bacterium, *Staphylococcus aureus*. The hydrogels showed excellent antimicrobial activity, finding that the most effective of the four hydrogels was the Ag-U gel in contrast to the Ag-A hydrogel. The Ag-pG was remarkably softer compared to the Ag-A gels in alkaline medium (storage moduli of Ag-A and Ag-pG were 3.06 and 0.03 kPa in the linear regime, respectively), whilst protonated adenine resulted in the formation of a white precipitate (Ag-pA). Along these lines, cytosine and guanine were also used to create new highly crystalline nanofibrous metallohydrogels with Zn(II) ions in an alkaline medium [13]. Gelation occurred mainly by coordination bonds and was accompanied by Zn hydroxide clusters formation through mixed chains of Zn(nucleobase) and Zn-(nucleobase)_2_ units, as the authors tentatively propose. These metallogels were synthesized by direct reaction between cytosine and guanine basified solutions with Zn(NO_3_)_2_·6H_2_O solutions, leading to hybrid flower-like superstructures. Rheological tests suggested that the elastic behavior was predominant over the viscous nature. This was authenticated by a more detailed study of the frequency sweep measurements, which clearly showed that in the frequency range of 0.05 to 100 s^−1^, storage modulus (G′) was always greater than the loss modulus (G″) (Table 1). These hybrid flower-like superstructures showed semiconducting properties and could be used as photocatalysts to degrade model organic pollutant dyes such as methylene blue.

To obtain a stable multifunctional MOG in the dark just in a single step through a spontaneous self-assembly process at room temperature is an objective that Neha Thakur et al. [38] have been achieved by mixing Ag(I) with commercial inosine 5′monophsphate (IMP) (Figure 9). This self-healable gel, formed by fibers of 8 nm in height measured by AFM and TEM, presents reversible gel–sol phase transitions in response to various external stimuli such as sulfate anions, acids, bases, and different salts. Like all MOGs that contain Ag(I), its presents interesting properties as antimicrobial, in this case, it is effective versus Gram-negative bacteria *E. Coli* and Gram-positive bacteria *S. aureus*.

Likewise, as most of those that contain Ag(I) ions, it can be photo-reduced by visible light, generating the corresponding silver nanoparticles, which in this case also act as a catalyst, reducing the model substrate p-nitrophenol to p-aminophenol. Rheological studies of this compound show that its mechanical properties are affected by both the concentration of Ag(I) and IMP (Table 1). An exciting feature of this gel is that it can separate oil from water and entrapment of polar impurities from a hydrophobic medium, which makes it attractive for potential applications in water treatment (Figure 10).

A novel and interesting application of metal–organic hydrogels based on the development of white light-emitting quantum dot (QDt)–gel systems using nucleobases was published in 2018 by B. Sharma et al. [39] In this work, supramolecular Cd(II)-thymine (Cd-T) and Cd(II)-uracil (Cd-U) metallohydrogels were used for the in situ growth of color-tunable CdS quantum dots. This advance has attracted much consideration due to the unique chemical properties of QDts and their technological applications, such as light-emitting devices, solar cells, energy scavenging, bio-sensing, and bioimaging [40,41,42,43]. These spontaneous hydrogels were formed through coordinative interaction with Cd(II) ions and these two nucleobases, at alkaline pH. As in the aforementioned work, the frequency sweep experiments were carried out in these gels with similar results (Table 1), showing that this type of solid-like behavior is characteristic of supramolecular gel systems. The introduction of Na_2_S to the synthesis allowed the obtaining of these QDts within the gels, with tunable emission color from blue to white to yellow just by varying the sulfide precursor concentration. The effect of temperature on the QDts emission was also studied, changing the reaction temperature during the synthesis. Using yellow emitting CdS-Cd-thymine hydrogel as a model system, the emission intensity for the QDts synthesized at 5 °C was maximum, while for the QDts synthesized at 60 °C was the lowest. At 25 °C, it showed intermediate emission intensity. This white light system was susceptible to quenching upon the addition of 0.476 mM of Fe^3+^ and Cu^2+^ ions, causing an emission quenching by 70% and 66% to the CdS-gel system dispersed in water respectively.

New attractive perspectives related to the manufacture of novel drug delivery composites have been recently obtained [43]. In particular, a recent study revealed that the stimuli-responsive nucleic acid-based polyacrylamide hydrogel could coat metal–organic frameworks (MOFs) nanoparticles for controlled drug release. A nanoMOF series of UiO-68 loaded with the antitumor doxorubicin was coated with a polyacrylamide copolymer and functionalized with two nuclei acid chains of DNA to form an ATP-sensitive hydrogel on the MOF surface. The release was produced when the hydrogel is disintegrated in the presence of ATP (overexpressed in cancer cells) via the formation of the ATP-aptamer complex. It seemed to be proportional to the concentration of the ATP trigger. These gelled materials reveal selectivity, which means that treatment with the other nucleotide triphosphates (TTP, GTP, and CTP) leads to an ineffective release of the load and effective cytotoxicity towards breast cancer cells in contrast with epithelial breast cells. This opens the door to the development of improved hydrogels that could act as a drug delivery mechanism with other types of stimuli such as pH, metal ions, lightning, or RNA biomarkers (Figure 11).

### 3.2. Amino Acids and Peptides as Gelators for the Creation of Novel Stimuli–Response, Self-Healing, Catalytic and Antibacterial Metal–Organic Hydrogels

Essential amino acids such as arginine, tyrosine, glycine, aspartic acid, histidine, phenylalanine, and their derivatives (Figure 12) are easily and naturally available in an affordable quantity-price ratio to prepare bulk materials using them as building blocks. Consequently, and due to their intrinsic properties of self-assembly, the structure and function of novel, bioinspired MOFs and CPs have been addressed. The human body uses amino acids to make peptides and proteins that help the body repair tissues. Taking as inspiration this self-healing ability, synthetic materials can be developed capable of recovering their original shape, structure, functionality, and properties after being damaged, either autonomously [44] or by external stimuli-response [35,45]. The main functional groups of the amino acids (amines and carboxylic groups) are great coordinating sites to different metal ions and the dynamism of these coordination bonds, associated with the equilibrium constant between the metal ion and the organic ligand, are key in the creation of self-healing metallogels.

Within the possible combinations à la carte, those that combine tyrosine-based amphiphiles (hydrophobic chains covalently linked with one or more hydrophilic amino acids) have aroused huge interest [45,46,47,48]. For instance, new interesting Ni(II) metal–organic hydrogels, synthesized in aqueous phosphate buffer solutions (pH 7.0 to 8.0), exhibited remarkable self-healing properties that can be tuned by modifying the tyrosine-based amphiphiles (B1–B4) chain length (Figure 13). Moreover, these metallogels showed multi-stimuli responsiveness towards various stimuli, including pH of the medium, temperature, mechanical forces, and external chemicals (Figure 14) [47]. In this study, all amphiphiles could not form any hydrogel in the presence of metal ions such as Zn(II), Cu(II), Co(II), Mn(II), Fe(II), and Hg(II), which excludes an anionic effect on gelation process. Therefore, the selectivity and specificity of the metallo-hydrogelation not only for tyrosine-based amphiphiles but also for the presence of Ni(II) can be emphasized.

The sol-gel transformations of Ni(II)-derived tyrosine hydrogels could be achieved when temperature decreases from melting point gel to room temperature, assisted by ultrasound. These hydrogels were also subjected to mechanical forces and then were transformed into solution. They were recovered when forces were removed, and the solution was kept without any disturbance at room temperature. To check their self-healing properties, gels were cut into two pieces with a razor, and put together again with a moderate press. These two parts fused into a continuous block after 25–30 min for all hydrogels. By modifying the alkyl chain length of these gelator molecules, this self-healability was successfully tuned, observing a stiffness increasing when the alkyl chain length increases. Rheological experiments confirmed the presence of a stable and rigid gel phase material in which G′ and G′′ did not vary significantly with the range of applied angular frequency and did not cross each other (G′ > G′′) throughout the experimental region (Table 2).

Hydrogels B1-B3 were subjected to a simple step strain experiment in several steps, increasing strain from 0.1% to 20%, constant strain at 20%, decreasing from 20% to 0.1% again, and constant strain at 0.1%. The recovery test results for B1, B2, and B3 gels were 80%, 80%, and 83% of their initial stiffness at the final interval showed.

Amphiphiles belong to an amazing class of low molecular weight gelators capable of having response against pH changes allowing to modulate sol-gel phase transition. This property can be exploited to propel the development of soft materials for encapsulation and the controlled release of biological molecules such as vitamins. Researchers such as S. Ray et al. tried to demonstrate this with a series of compounds made up with phenylalanine-based bolaamphiphiles (named 1–4), being 1 the only one able to form hydrogels with divalent metal salts (such as MnCl_2_, CoCl_2_, CuSO_4_, and NiCl_2_) thanks to it containing a centrally located oligomethylene group (Figure 15) [46].

The critical point is the substantial solubility variation of the bolaamphiphiles with the pH. Therefore, once they are dissolved with NaOH solution, and after pH was adjusted between 6.5 and 7.2, metals salts were added to the previous solution (2:1 bolaamphiphiles:metal stoichiometry), obtaining four hydrogels. The gel–sol transition at higher pH was not fully understood. At the higher pH values, the ionization of carboxylic acid moieties could be helping the sol-gel transition. However, at lower pH (<6.5) one of both carboxylic acid groups is protonated which decreases the solubility and, therefore, the tendency of the bolaamphiphile to form gels. The pH-responsive metallohydrogels were allowed to adsorb different dyes from water, indicating their possible applications in wastewater treatment. One of these metallohydrogels also entrapped vitamin B12 molecules (Figure 16) and they could be released when the pH medium changes, suggesting a possible use of this gel as a carrier of other biological compounds. Using histidine-based bolaamphiphiles instead of phenylalanine-based ones, selective hydrogels can be formed with Cu(II) ions over other metal ions. This acid-tolerable hydrogel suffers drastic morphological changes, in which proton and copper ions, respectively, can trigger the self-assembly into single-wall nanotubes or single-molecule thick fibers [49].

Vittal et al. [50] have reported the formation of a 1D coordination polymer gel (Figure 17), which aggregate to form 3D network fibrous nanostructures through non-covalent interactions to catch water molecules. This gel is formed with *N*-(7-hydroxyl-4-methyl-8-coumarinyl)-glycine (H_2_mugly) and Zn(II) in a basic aqueous solution and is pH-responsive. If the solution is acidified to pH 2, the gel converts into a clear colorless solution, but it can recover its gel phase when the pH is raised to 8. In addition, this hydrogel exhibits a strong blue emission which decreases at lower pHs. The ligand protonation seems to be a reasonable explanation for these behaviors at acidic pH values, leading to the disruption of complexation and the quenching of the fluorescence. Dynamic oscillation and steady shear measurements were carried out to understand the rheological properties (Table 2) of the 3D structure of hydrogel. They also studied in detail the nature of the macromolecular assembly through the correlation known as the Cox–Merz rule. Interestingly, the hydrogel violated the Cox–Merz rule [51], contrasting with macromolecules that interact solely via topological entanglements. Dynamic viscosity measurements were around two orders of magnitude higher than steady shear viscosity from small deformation oscillatory at equivalent deformation rates. Those results seemed to support a weak network that remains intact under low amplitude oscillation but is disrupted by continuous shear. In this line, these authors have also reported a pH and mechano-responsive coordinationpolymeric gel by reaction of Mg(II) aqueous solution with the basic aqueous solution of N-(7-hydroxyl-4-methyl-8-coumarinyl)-alanine (Figure 18) [52]. Herein again, the cause of the gelation process might be connected to the formation of 3D nanostructures through non-covalent interactions due to the self-aggregation of 1D coordination polymers. Morphological studies of the freeze-dried hydrogel showed a fibrillar network from ribbons whilst UV–vis absorption studies indicated the hydrogel exhibited a typical π–π* transition. Once the gel is formed, a longer lifetime enhanced its fluorescence intensity, improving the properties of the Zn-hydrogel as mentioned above. Thanks to the information achieved about its supramolecular nature through a detailed study of its mechanical and rheological properties, together with its biocompatibility, this Mg(II) hydrogel might be useful in biomedical applications.

Wang et al. [53,54,55] have synthesized a serial of tryptophan and phenylalanine derivatives with specific gelation properties induced by metal ion (Figure 19). The gelator 2-((4-(hydroxymethyl)benzyl)amino)-3-(1H-indol-3-yl) propanoic acid (HAIP) can selectively interact with Pb(II) ions to form stable gels, while the PT-gelator forms a gel with Co(II) ions, with both gelators belonging to the tryptophan derivatives group. The regioisomers of phenylalanine derivatives (PF-gelator) specifically responds to Ni(II) ions. All hydrogels have pH-responsive and show utterly different gelling abilities due to the various positions of the pyridine nitrogen atoms or by carboxyl and amino groups in HAIP, responsible for the Pb(II) ions coordination. More recently, these authors tried the possibility of selectively constructing a new La^3+^-metallohydrogel by modifying the above gelator. They introduced an imidazole group with two N atoms in the phenylalanine derivative (*N*-(1H-imidazol-4-yl)methylidene-l-phenylalanine) to provide the necessary multiple coordination sites for La^3+^ ions and H-bonds for the hydrogels [56].

Recently, Vadivel Sasikala et al. [57] designed multifunctional gel based on arginine [58], 1,3,5-benzene tricarboxlic acid (BTC) and Co(II) with tunable morphology, caused by modifying pH. These structural changes are probably due reorganization in the metal ion coordination mode at different pH, leading to 1D microfibers (pH 5), 2D micro sheets (pH 7), 3D rugged (pH 8) and mixture of different morphologies (pH 9). Its mechanical stability was studied showing a catastrophic disruption of the 3D net beyond 30% of strain (Table 2). This gel of mesoporous nature (407 m^2^ g^−1^) shows a distinct fluorescence that quenches with the addition of copper(II), offering small detection limits (0.0547 × 10^−7^ M) for copper ions detection. This chemosensing gel displayed a high antibacterial activity against *S. Coccus* and *S. Marcescens* breaking the cell wall of these pathogens.

However, lanthanides compounds are especially attractive as luminescent materials due to their sole optical properties, such as narrow line-like emission, significant Stokes shifts, long lifetimes and high luminescent efficiency. Phenylalanine (Phe) has also been employed as sensitizer to enhance the lanthanide ions luminescence via an “antenna effect” and avoid water molecules quenching the luminescence of lanthanide ions. Its reaction with Tb(III) or Eu(III) using water as solvent led to long lifetimes of environment-friendly metallohydrogels since water molecules from the coordination sphere could be replaced by Phe molecules resulting in strong luminescence (green emission in case of Phe-Tb and red emission for Phe-Eu), with unclear solid-state structure (Figure 20) [59].

The luminescent spectra of these hydrogels exhibit reversible gel-sol transition along with the reversible luminescent ON/OFF properties. The dynamic coordination bonds made them pH stimuli respond, and the shearing thinning properties convert them into useful luminescent inks for anti-counterfeiting.

On the other hand, silver-related materials are well-known antimicrobial drugs specially to treat burn wounds due to their biocidal effects. Effective antibacterial Ag(I)-glutathione bio-coordination polymer hydrogel has also been synthesized by Y. Liu et al. [60] The glutathione (GSH) is a non-protein tripeptide formed by three amino acids: glutamate, cysteine, and glycine, with antioxidant properties for cells helping protect them from reactive oxygen species such as free radicals and peroxides. Because it contains one sulfhydryl and two carboxyl groups, among others, it is a perfect candidate to create coordination polymers. The Ag(I)-GSH hydrogel was synthesized by mixing equimolar AgClO_4_·H_2_O and GSH in a solution with a final pH around 5.2. After several hours, the hydrogel was formed. However, the amount of Ag(I) ions released and required for its application was impaired and unfavorable due to its high solubility in water. To solve this problem, solubility in water was decreased, introducing biocompatible Ca(II) ions into the Ag(I)-GSH hydrogel system to make it cross-linked. Comparing powder X-ray patterns of both xerogel and precipitate, and undertaking a detailed analysis of them, the authors proposed that Ag(I) ions and sulfur atoms of GSH formed two-dimensional slabs through Ag–S bonds with both Ag and S in a three-coordinate mode. At the same time, the GSH carboxylate groups in adjacent layers are connected through ionic bridges of Ca(II) ions. Rheological measurement of the strengthened hydrogel through cross-linking presented a storage modulus (G′) of up to 2 × 10^4^ Pascal, two magnitude orders more than in the gel before cross-linking with Ca(II) ions [61,62,63,64,65,66,67,68]. However, the cross-linked hydrogel is broken when the strain is larger than 2% (Table 2).

In 2015, Yuanyuan et al. [61] developed a glutathione-derived peptide ligand whose N-terminal end is protected with a fluorenylmethoxycarbony group (Fmoc-GCE-OH, G = glycine, C = cysteine, and E = glutamic). This ligand was used in the construction of self-assembled peptide structures. In this case, the self-assembly generated by the gel into nanofibres occurs thanks to the coordination through Ag(I) ions due to its high affinity for the cysteine residues contained in the ligand structure, and also thanks to the π–π stacking of fluorenyl groups. Once the coordination is produced and the self-assembly is fixed, silver nanoparticles can be obtained along the already formed nanofibers, using reducing agents or light (Figure 21).

The nanoparticles were obtained by the addition of reducing agents such as sodium borohydride or by irradiation with light, observing a color change in the solution to dark brown and giving rise to a colloid. The TEM study confirmed the role of silver ions in the self-assembly process of the nanofibers, which are destroyed when the nanoparticles are formed.

On the other hand, the Fmoc-GCE/Ag^+^ solution presents a sol–gel behavior depending on the NaOH concentration in the medium, forming the gel at 16mM with rheological properties characteristic of these materials and indicating an elastic domain (Table 2). In addition to the concentration of NaOH, it also presents this type of stimulus-response in the presence of pyridine and melamine.

The AgNPs growing along the Fmoc-GCE nanofibers exhibiting antibacterial properties to both Gram-negative (*E. Coli*) and Gram-positive (*S. aureus*) bacteria. Finally, its catalytic activity was evaluated, producing the complete degradation of methyl orange in a few minutes. They also tested its bactericidal effect against *E. Coli* and *S. cereus* and found that the minimum inhibitory concentration of the compound is 0.14 mM.

Following the line on how silver ions can affect the processes of peptide self-assembly and gel formation, a class of short aliphatic peptides has been designed in order to take advantage of their structural simplicity, low synthesis cost, and biocompatibility. Specifically, a histidine-derived peptide called IH6 was developed in 2019 [62], with which the silver activity was evaluated, not only in the assembly process but also the controlled release of silver and its biological properties once the Ag-peptide matrix is formed. By performing a detailed study using circular dichroism spectroscopy, they investigated how the conformation of IH6 would be affected by its response to silver ions, which changed from a mostly random coil to a β-sheet structure, increasing the mechanical stress of the gel (Table 2, Figure 22). Thanks to the good results obtained in the copper release assays, they proceeded to study this compound against *E. Coli*, *P. Aeruginosa*, and *S. aureus*. It was observed that 1 mM was the minimum amount of silver ions in the hydrogel to cause a total suppression of the growth of the strains. Based on these results, the authors further advanced the study of this gel as a possible candidate wound-dressing material. They found that this compound can provide selective killing of bacteria previously inoculated into human cells with *E. Coli*.

Recently, different biometallohydrogels in the form of nanofibers with interesting mechanical properties based on the self-assembly and local mineralization of Ag^+^-Fmoc-amino acids (*N*-(fluorenyl-9-methoxycarbonyl) (Fmoc)-modified amino acids), have been developed. These Fmoc-amino acids metallohydrogels act as precursors to produce AgNPs [63]. The high directionality and anchoring ability of the coordination interactions allow silver ions to bind within the hydrogel nanofibers. These materials show advantages compared to traditional gels formed by peptide-containing nanoparticles, as they can be used in in vivo assays for localized drug delivery and controlled antimicrobial amino acids and AgNPs release resulting in reduced drug dosage and toxicity, improved bioavailability, prolonged drug effect, and adjustable mechanical strength. This material can act as a broad-spectrum antimicrobial since it is able to trigger the bactericidal effect against *E. Coli* and *S. aureus*. This effect is produced by the interaction of the biometallohydrogel with the cell membranes in cultured cells and mice (Figure 23), causing detachment of the plasma membrane and leakage of the cytoplasm and, therefore, cell death.

With the appropriate choice of transition metals, metal–organic gels [64,65] with interesting catalytic activities can be designed. An example using Cu(II) and amino acids have been recently published by Hao Qiu et al. [25] who developed one coordination polymer hydrogel-based artificial enzyme with peroxidase-like activity for combating bacteria and accelerating wound healing (Figure 24).

This copper(II)-aspartic acid gel was previously synthesized by D. Maspoch’s [66] group in 2009 by the reaction of a Cu(NO_3_)_2_, L- aspartic acid, and NaOH in a water solution. After, H. Qiu et al. evaluated the catalytic activity of the hydrogel by oxidation of a chromogenic substrate, 3,3′,5,5′-Tetramethylbenzidine (TMB), in the presence of H_2_O_2_. They monitored the time-dependent absorbance in a spectrophotometer of UV–vis at 652 nm, and a strong absorption was observed when the hydrogel was present, indicating great peroxidase-like activity. Of course, it was previously verified that the same effect was produced by the starting reagents separately, indicating zero catalytic activity in both cases. Subsequently, the generation of hydroxyl radicals (·OH) was also verified by the hydrogel when H_2_O_2_ is decomposed, in which terephthalic acid forms the 2-hydroxy terephthalic acid, producing a remarkable fluorescence at 435 nm after incubation for 12 h. In vitro assays showed that this hydrogel resulted effective against both drug-resistant Gram-positive bacteria and Gram-negative bacteria. Additionally, they demonstrated an enhancement in wound healing by stimulating collagen deposition and angiogenesis because copper ions could be slowly released from the hydrogel.

Another metal–organic hydrogel, with catalytic activity towards fixation of SO_2_ and CO_2_ with epoxides, has been obtained by the combination of Na_2_HL {H_3_L = 2-{(3,5-di-tert-butyl-2-hydroxybenzyl)amino succinic acid with Cu(II) (called Cu-MOG) [11]. When stoichiometric aqueous solutions of the ligand and copper chloride were mixed in a glass vial at room temperature, a dark green hydrogel was instantly obtained. The authors theorized that copper chloride’s reaction with the ligand’s disodium salt would afford a 1:1 copper complex with free carboxylate groups, leading to the hydrogel formation due to the extensive hydrogen-bonding interactions between these carboxylate groups and water. This hydrogel presents a quick thixotropic behavior when an external force is applied (the longer the fluid is subjected to shear stresses, the greater its viscosity decreases) and multi-stimuli responsive nature (Figure 25). The xerogel of this coordination compound can be utilized as a catalyst for the chemical conversion of SO_2_ to cyclic 1,3,2-dioxathiolane-2-oxides with high diastereoselectivity and the transformation of CO_2_ to carbonates by reaction with the epoxides under standard conditions. Hydrogel mechanical properties were measured by amplitude sweep which found the G′ was larger than G′′ over the entire region of frequency, and the frequency-invariant characteristic confirmed the viscoelastic nature of the metallohydrogel (Table 2). The thixotropic behavior was tested through the step strain experiment. In this test, increasing and decreasing cycles of a percentage strain were applied to cause the gel-to-sol transformation and vice versa, checking that the recovery of the MOG was almost 100%.

C. Xu et al. [67] found that the peptide Nap-GFFYGGGHGRGD could self-assemble into a gel upon binding to Zn(II). It could frustrate microbial growth due to the antibacterial activity of Zn(II) (Table 2 and Figure 26). The polypeptide sequence was synthesized following some principles: (i) the motif Nap-GFFY was used to facilitate the work of forming gels to short peptides containing it; (ii) peptide sequence GGH (Gly-Gly-His) was added as the metal-binding site because of its background; (iii) the RGD sequence (tripeptide Arg-Gly-Asp) was included to try to increase the hydrophilicity and improve the biocompatibility. Upon proving several conditions, the researchers optimized the gel formation establishing that the minimum equivalent of Zn(II) to form hydrogel was 0.2 wt% in phosphate-buffered saline (PBS), and the peptide:Zn ion ratio was fixed at 1:1, being the minimum concentration of peptide to form hydrogel 0.3 wt%. This procedure was tried with other metal ions: Ca^2+^, Mg^2+^, Ba^2+^, Mn^2+^ y Sr^2+^, but only hydrogel formation happens with Ca^2+^, Mg^2+^. They also demonstrated that the gel formation did not happen by removing the GGH sequence from the peptide formula. Interestingly, TEM images and the rheology results showed that the higher zinc concentration is related to the fibers being stronger, leading to hydrogels with an improved mechanical strength because of the larger force-bearing capacity of the fibers. In vitro assays were carried out in the *E. Coli* cell cultures in which hydrogel can inhibit the growth of this bacteria inside and on the hydrogel surface and its antibacterial activity depend mainly on the amount of zinc ions. However, although the ions concentration is critical to improving antibacterial activity, the metal presence is only effective if the ions are encapsulated inside the gel and not as free ions in solution.

Later in 2019, L. Zeng et al. [68] synthesized two self-healing hydrogels based on the coordination of a Gly-Gly-His tripeptide (GGH) and a Gly-His-His-Pro-His polypeptide chain (GHHPH) as ligands with Zn(II) as the metal center (Table 2 and Figure 27). They investigated the possibility of mechanical resistance enhancement by increasing the coordination binding sites without sacrificing the stretching ability and self-healing. The different polypeptide chains and acrylamide solution were first mixed in MilliQ water for their synthesis and degassed with an ultrasonic bath under argon atmosphere. Ammonium persulfate was then added as a photoinitiator to cause polymerization by irradiating with UV light. The resulting hydrogel was immersed for 24 h in Tris buffer at pH 7.6, which additionally contains KCl and different concentrations of ZnCl_2_ to provoke the metal coordination. They found that the hydrogels based on GHHPH had higher hardness and Young’s modulus values than those synthesized with GGH. By increasing the peptide concentrations in the synthesis, the authors demonstrated that the coordination number of the cross-linkers plays a fundamental role achieving to synthesize gels with greater stiffness, faster self-healing rate, and higher stretching ability and toughness values (867 and 1300 kJ/m^3^ for GGH and GHHPH, respectively).

## 4. Aerogels Based on Coordination Bonds with Biological Interest Ligands

### Amino Acids and Nucleobases as Building Blocks to Generate Novel Composite/Hybrid Metal–Organic Aerogels Useful for Energy and Bioanalytical Applications

Numerous applications of composite aerogels have been reviewed [69,70,71], including catalysis, separation, adsorption, energy conversion, and storage devices such as batteries and supercapacitors. Graphene oxide (GO) aerogels have recently been developed to overcome the lack of electrical properties when an initial GO macrostructure is assembled to form 3D graphene networks [70]. Thus, the goal is to form unions between graphene sheets to reinforce the assembly and provide conductive interconnections between the individual sheets [72]. On the other hand, aerogels are usually limited for their practical application due to their weakness and brittleness. To solve that, these materials can be doped or mixed with other chemical agents [73] to reinforce those properties. For example, dopamine (DOPA) can be considered as a reductant and a surface modifying reagent for GO. Also, polyaspartic acid, with –OH or –COOH functional groups, can interact with GO sheets via covalent or non-covalent cross-linking. This combination was carried out by Wang B. et al. [74], who manufactured a novel aerogel based on GO and amino acid derived from polyaspartic acid, modified with DOPA, termed PAAD. Once PAAD and GO were synthesized, both compounds were mixed in water and subjected to hydrothermal conditions to obtain a hydrogel (Figure 28) that was incubated with a FeCl_3_ (in excess) solution at different pHs (being the optimal pH 9) for 12 h forming a physically cross-linked network. The water-soluble PAAD could efficiently react with the GO by strong hydrogen bonds with the oxygen functional groups on the surface of GO sheets. The catechol moiety from DOPA formed a strong and pH-responsive reversible coordination bond with Fe(III), acting as the secondary cross-linker to further strengthen the PAAD/GO hydrogel by coordinating Fe(III) and carboxylic acid groups and catechol groups. The aerogel was obtained by freeze-drying the wet gel and, after, was reduced at 200 °C for 24 h by hydrothermal reaction method. The final PAAD-rGO-Fe aerogel provided high conductivity, high porosity and surface area (Table 3), low density, and robust mechanical properties, with reversible compressibility. This novel aerogel exhibited high specific capacitance (276.4 F g^−1^ at 0.5 A g^−1^) and long cycle life for energy storage (capacitance retention of 88.2% after 5000 cycles), making into a promising supercapacitor electrode due to the reversible redox reaction quinone (Q)/hydroquinone (QH) (QH_2_
↔ Q + 2H^+^ + 2e^−^) provided by the catechol groups.

Following with the manufacture of composite materials to reinforce properties, the combination of two types of fibrous building blocks allowed the fabrication of a novel biocompatibility composite aerogel by the bottom-up approach and freeze-drying technique. This aerogel was obtained from the assembly of cellulose fibers with a MOF, in which Cu(II) ions are coordinated to L-cystine ligands, and was coated with polyallylamine hydrochloride (PAH) allowing their covalent linkage to TEMPO-oxidized cellulose by means of EDC/NHS (N-hydroxysuccinimide=NHS; N-(3-dimethylaminopropyl)-N′-ethylcarbodiimide hydrochloride=EDC). This lead to blue-colored nanocrystalline structures (4% of L-cysteine Cu(II)-MOF) [75]. To avoid the partial release of the MOF fibers from the cellulose matrix, the blue foam was prepared using the ice-templating method from 3% *w*/*v* suspensions, freezing the samples at −20 °C and then producing the sublimation of ice by lyophilization. This material was proved to decompose low molecular weight S-nitrosothiols (RSNOs) similar to those existing in blood, to release nitric oxide (NO) (Figure 29) as an antimicrobial agent, with the idea of its application as implants or wound-dressing materials. Unfortunately, the quantities of generated NO (around 360 nM) were insufficient to display a bactericidal effect on the studied bacteria *E. coli* and *S. epidermidis*, in the presence of a physiological concentration of RSNOs. Despite this, cytotoxicity studies revealed this foam is friendly enough for further biological studies.

The fact that the aerogel formation can provide materials with a certain porosity that previously they did not have is outstanding material engineering. On the other hand, one of the least-studied chemical systems that can be applied to obtain new aerogels is based on CPs. As we have already mentioned, there are many fewer works in which the aerogel structure is accurately known. Interestingly, V. G. Vegas et al. [76] have overcome this issue by designing a 1D coordination polymer (named 1) functionalized with uracil residues with molecular recognition. They combined Cu(OAc)_2_, uracil-1-acetic acid (UAcOH) and 4,4′-bipyridine (4,4′-bipy) in a stoichiometry 1:2:1 in water at normal conditions (pH= 5.2), to synthesize a 1D CP double chain based on a [Cu_2_(μ-CH_3_COO)_2_(μ-4,4′-bipy)_2_] core adorn with UAcO moieties along both sides of the double chains. The fine-tuning of the synthetic parameters, in which the addition of 0.07 mL of CH_3_COOH was essential, allowed the formation of a purple water colloid (1n) composed of nanoribbons of 1. This colloid was turned into a hydrogel upon sonicating the 1n for 20 min and leaving the sample 24 h at room temperature (Figure 30). Finally, the supercritical CO_2_ drying of the hydrogel led to the 1n-aerogel. The dimensions of the ribbons were measured by atomic force microscopy (AFM) and scanning electron microscopy (SEM), which showed a broad dispersion in thickness with values between 3 to 80 nm, and with average values of 28 ± 10 nm for 1n and 61 ± 18 nm for 1n-Aerogel. This material had a specific surface area of 21 m^2^ g^−1^ (Table 3), and it can be ascribed to the ribbon-like shape of the crystals comprising the aerogel. As a proof of concept, a stationary phase for high-performance liquid chromatography (HPLC) was made up with this MOA thanks to its meso/macroporous nature, nanosize, and the highly selective supramolecular interaction caused by the presence of nucleobases.

## 5. Conclusions

This overview has enabled us to highlight new bioinspired colloids, gels, and aerogels based on coordination bonds as a recent research area that has been emerging in recent years by allowing the creation of new biological materials with interesting advantages and novel applications in biomedicine and bioanalytical techniques (Table 4).

The dynamism of metal–organic biomolecule interaction in these soft materials has not only resulted in controllable nanostructures but also induced multi-stimuli responsive behaviors. Therefore, nucleotides or nucleobases have been demonstrated to be useful in the manufacture of nanocarriers metal–organic colloids/gels with selective molecular recognition, entrapping guest molecules, and pH response, acting also as drug-delivery materials; or even amino acids-based hydrogels have shown bactericide and wound-healing properties. Herein, the selection of metal ions such as Zn(II), Ag(I), Ln(III), Co(II) or Cu(II) is one of the most important factors in the tailoring of new metal–organic gels and aerogels with antimicrobial and antibacterial qualities. Drug delivery is useful as artificial enzymes due to their catalytic activity. Some of these reported works show the improvement of processability and handling regarding powdery CPs. Indeed, their synthesis in many cases leads to a reduction in size, resulting in their synergy with nanotechnology and the launching of new technological advances such as 3D printing, light-emitting devices, solar cells, anti-counterfeiting, energy scavenging, biosensing, and bioimaging. The work also shows how much remains to be done in relation to new possible applications, such as in energy, where the emerging area of bio-batteries, biological fuel cells or microbial fuel cells, and bio-inspired ligands such as enzymes used together with metals have not yet been explored.

Despite such astonishing applications, there are challenging issues that still need to overcome overall those related to structure-property correlation of CPGs and their characterization. The development and use of new characterization techniques have become a fundamental strategy, as a single-crystal structure determination is not possible here. All these hybrid materials and their synthesizing strategies are broadening the way forward for the rational optimization of CPs and MOFs with the enhancement of stability, easy manufacturing, accessibility, and flexibility, which will emerge in potential applications shortly.

## Figures and Tables

**Figure 1 nanomaterials-11-01865-f001:**
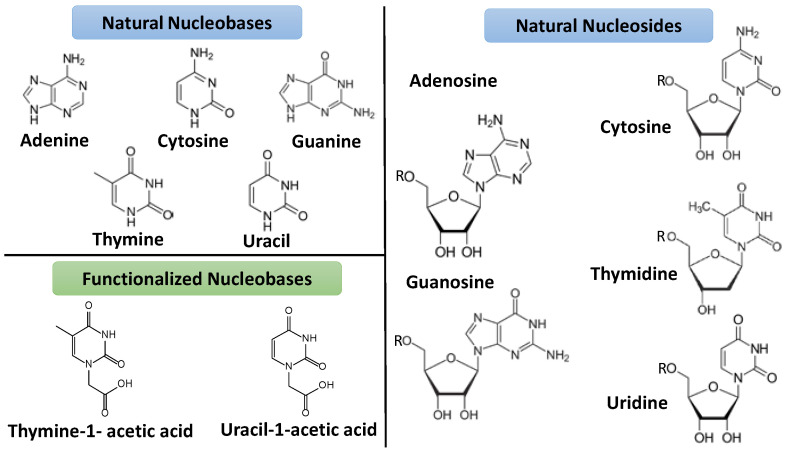
Structure of natural nucleobases and nucleotides (where R = mono-, bi- or triphosphate groups) that form DNA structure and some artificial or functionalized nucleobases.

**Figure 2 nanomaterials-11-01865-f002:**
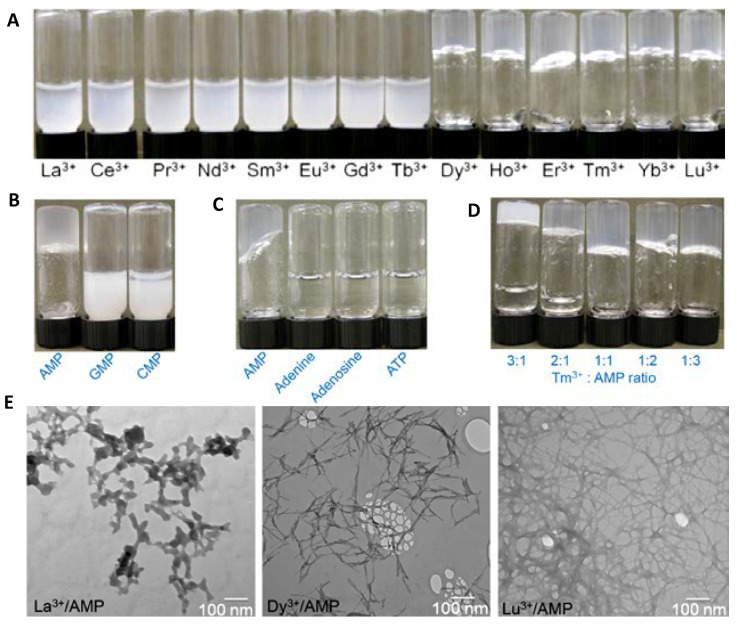
Photographs of the colloid suspensions and metal–organic gels formed by mixing various Ln(III) (**A**). Images of the formed products between Dy(III) with different (**B**) nucleotides or (**C**) adenine and its derivatives and (**D**) changing the ratios while keeping the sum of Tm(III) and adenosine monophosphate (AMP) concentrations. Transmission electron microscopy (TEM) images of the coordination compounds formed by mixing AMP with La(III), Dy(III), and Lu(III) after overnight incubation (**E**). Adapted from reference [25] with permission, copyright 2018 American Chemical Society.

**Figure 3 nanomaterials-11-01865-f003:**
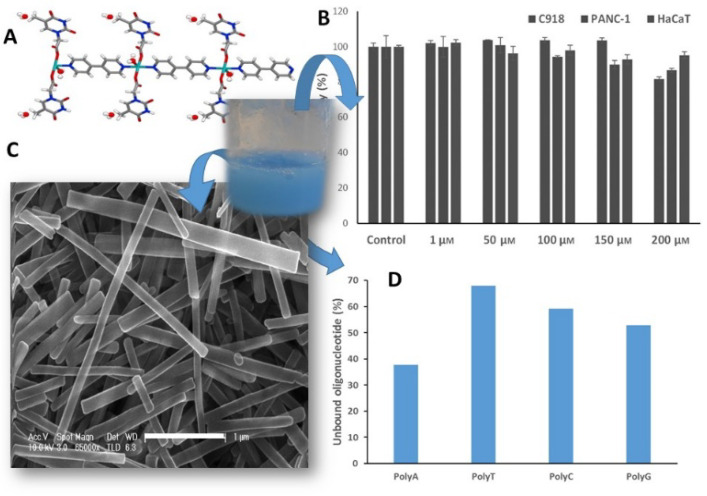
One-dimensional Cu(II) coordination polymer (**A**) as colloidal suspension of nanometric dimensions (**C**), with low toxicity versus different cancer cell lines (C918, PANC-1, and HaCaT) (**B**) and with selectivity versus adenine oligonucleotide (**D**). Adapted from reference [18] with permission, copyright 2019 John Wiley and Sons, Inc.

**Figure 4 nanomaterials-11-01865-f004:**
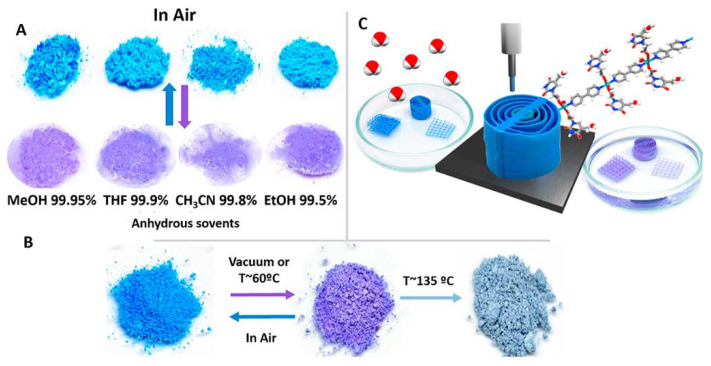
Photos of the 1D Cu(II) antiferromagnetic coordination polymer with thymine-1-acetate powders after soaking for 1–5 min. in different dry organic solvents (**A**). Conversion process mediated by either temperature or vacuum (**B**). Schematic illustration of the 3D printing material with color change in organic solvents (**C**). Adapted from reference [17] with permission, copyright 2019 John Wiley and Sons, Inc.

**Figure 5 nanomaterials-11-01865-f005:**
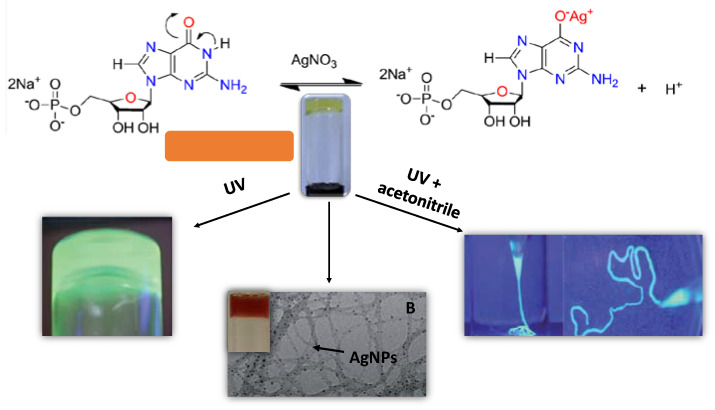
1:1 Ag-GMP (guanosine monophosphate) hydrogel exposed to ultraviolet (UV) light (**A**); when it is synthesized in stoichiometric 1:2/GMP:Ag and in the presence of cytochrome C (**B**); 1:1 Ag-GMP hydrogel after injection into acetonitrile and exposed to UV light (**C**). Adapted from reference [24], with permission, copyright 2011 Royal Society of Chemistry.

**Figure 6 nanomaterials-11-01865-f006:**
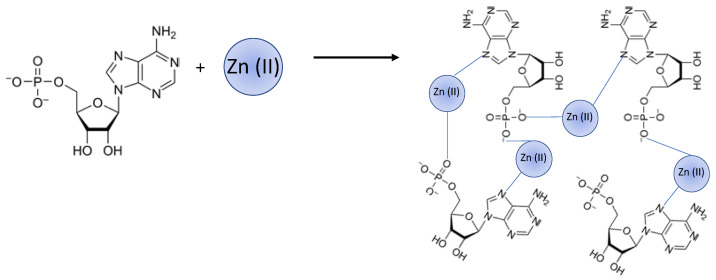
Scheme of Zn(II) coordination mode with AMP and the proposed structure for the Zn-AMP gels.

**Figure 7 nanomaterials-11-01865-f007:**
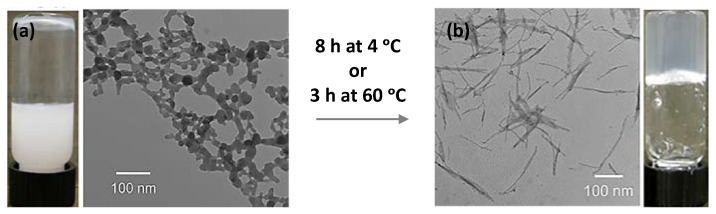
Transmission electron microscope (TEM) images of AMP with Dy(III) coordination compound at time 0 h (**a**), after 8 h or 3 h at 4 °C and 60 °C, respectively (**b**). Adapted from reference [25] with permission, copyright 2018 American Chemical Society.

**Figure 8 nanomaterials-11-01865-f008:**
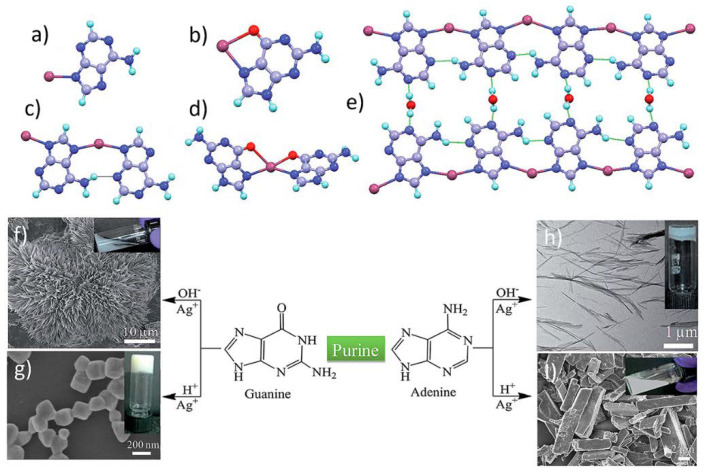
Monomeric and dimeric units of Ag-A (**a**,**c**) and Ag-G. (**b**,**d**). Proposed structure of the polymeric Ag-A hydrogel (**e**). Schematic representation for forming coordination polymer superstructures with purine nucleobases and field-emission scanning electron microscope (FESEM) images of Ag-G and Ag-A, as precipitate (**f**–**i**) and gel (**g**,**h**). Adapted from reference [37].

**Figure 9 nanomaterials-11-01865-f009:**
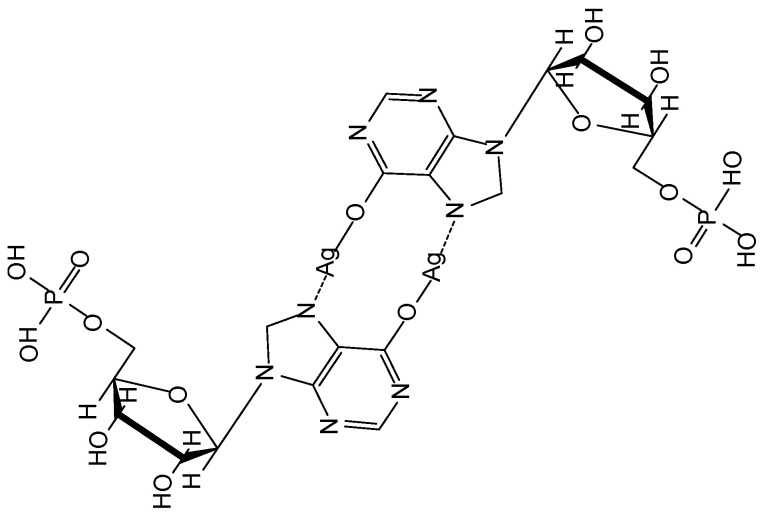
Chemical structure of Ag-IMP metalorganic gel.

**Figure 10 nanomaterials-11-01865-f010:**
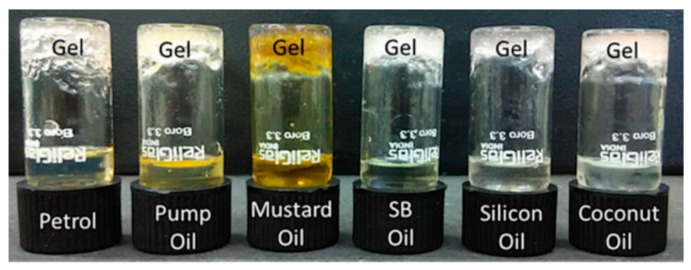
Images of phase-selective separation of water from different water-oil mixtures by the Ag-IMP gel. Adapted from reference [38] with permission, copyright 2018 American Chemical Society.

**Figure 11 nanomaterials-11-01865-f011:**
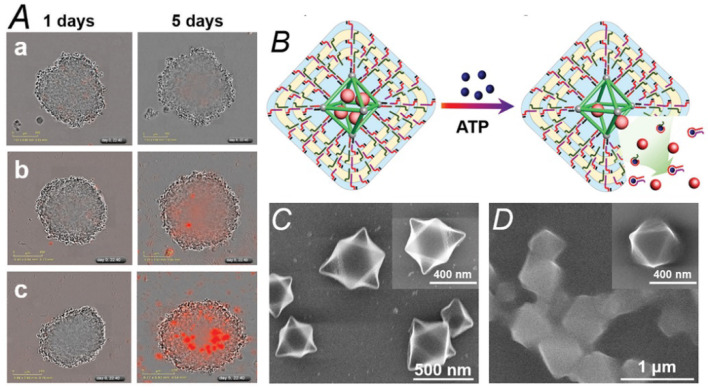
(**A**), Different stages of the cell aggregates after time intervals of 24 and 120 h: (**a**) cells treated with only hydrogel-coated nanoMOFs (metal–organic frameworks), (**b**) cells treated with nanoMOFs together with two acid nucleic chains loaded with doxorubicin and (**c**) cells treated with hydrogel-coated nanoMOFs loaded with doxorubicin. (**B**), Representation of the hydrogel-coated nanoMOFs unlocking mechanism and release the load via the formation of ATP–aptamer complexes. (**C**), Scanning electron microscopy (SEM) images of the nucleic acid-functionalized UiO-68 NMOFs before the deposition of the hydrogel and (**D**), SEM images of the hydrogel-coated nanoMOFs. Adapted from reference [43] with permission, copyright 2017 John Wiley and Sons, Inc.

**Figure 12 nanomaterials-11-01865-f012:**
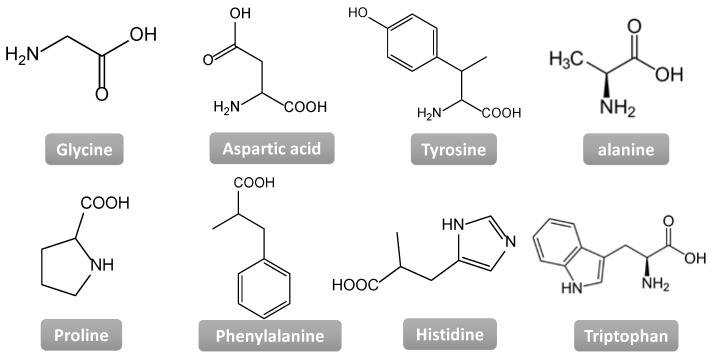
Some of the essential amino acid structures, used as building blocks to create metal–organic hydrogels.

**Figure 13 nanomaterials-11-01865-f013:**
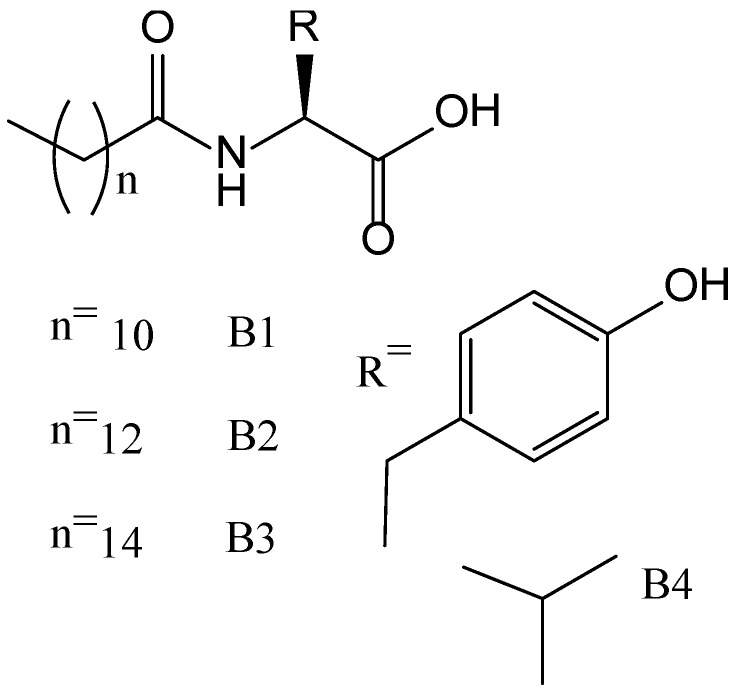
Chemical structures of tyrosine-based amphiphiles B1–B4.

**Figure 14 nanomaterials-11-01865-f014:**
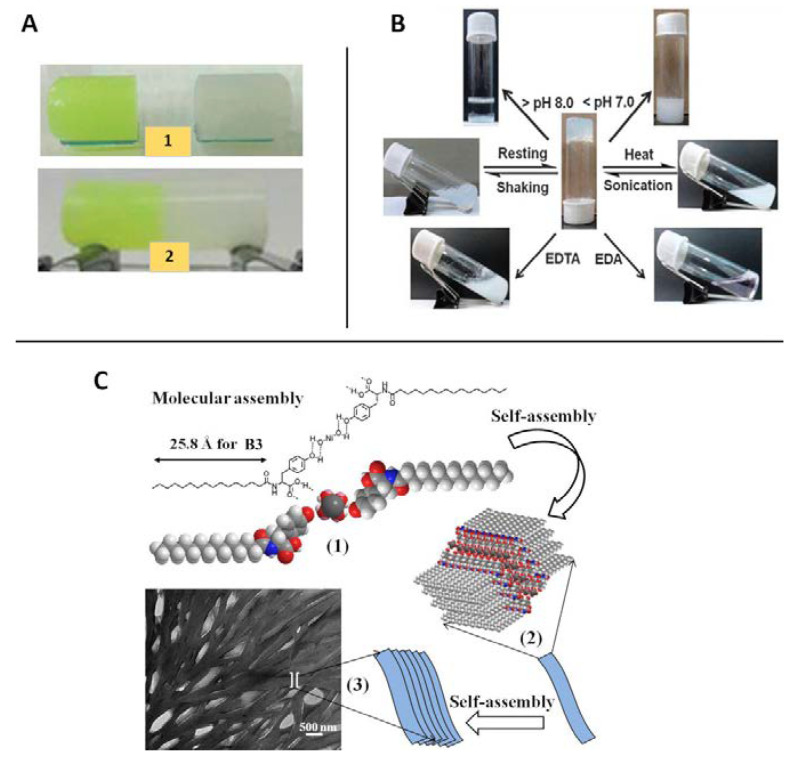
Ni(II)-derived tyrosine hydrogel photographs showing its self-healing property (**A**); scheme of the sol-gel transformations (**B**), and the alkyl chain length effect (**C**). Adapted from reference [47].

**Figure 15 nanomaterials-11-01865-f015:**
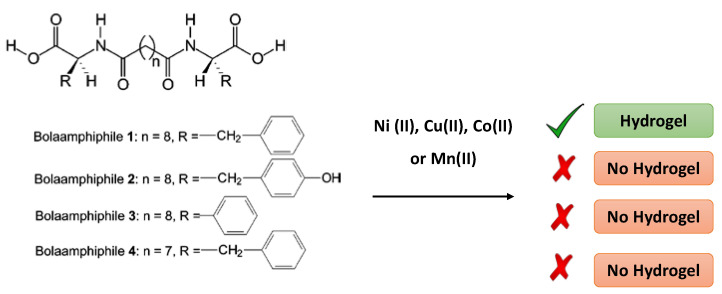
The bolaamphiphiles series (**1**–**4**) that form hydrogels with MnCl_2_, CoCl_2_, NiCl_2_ or CuSO_4_. Adapted from reference [46] with permission, copyright 2007 American Chemical Society.

**Figure 16 nanomaterials-11-01865-f016:**
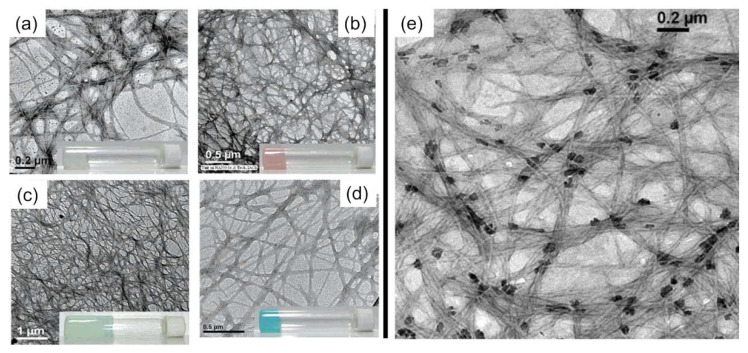
TEM images of bolaamphiphile 1-based hydrogels at pH 6.5 (**a**) with MnCl_2_, (**b**) CoCl_2_, (**c**) NiCl_2_, and (**d**) CuSO_4_. TEM image of the bolaamphiphile 1-Co metallohydrogel with vitamin B12 trapped within the gel nanofiber network (**e**). Adapted from reference [46] with permission, copyright 2007 American Chemical Society.

**Figure 17 nanomaterials-11-01865-f017:**
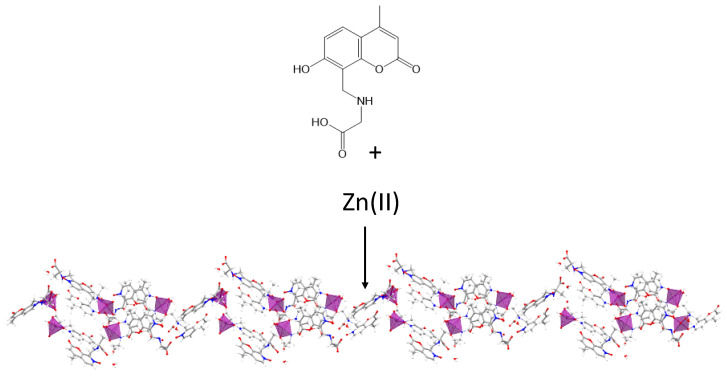
Fragment of the 1D network of the hydrogel structure.

**Figure 18 nanomaterials-11-01865-f018:**
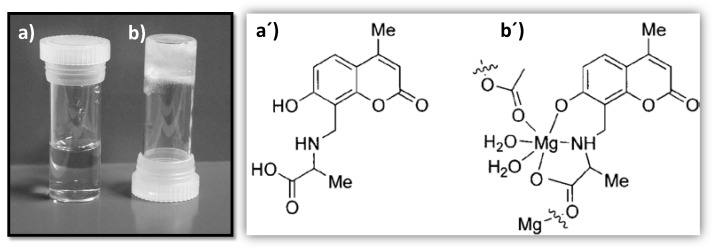
Photograph of the ligand solution (**a**) and its structure (**a′**); and photograph of the mixture coordination polymer Mg-alanine derivative as forming a hydrogel (**b**) and its proposed structure (**b′**). Adapted from reference [52] with permission, copyright 2008 John Wiley and Sons, Inc.

**Figure 19 nanomaterials-11-01865-f019:**
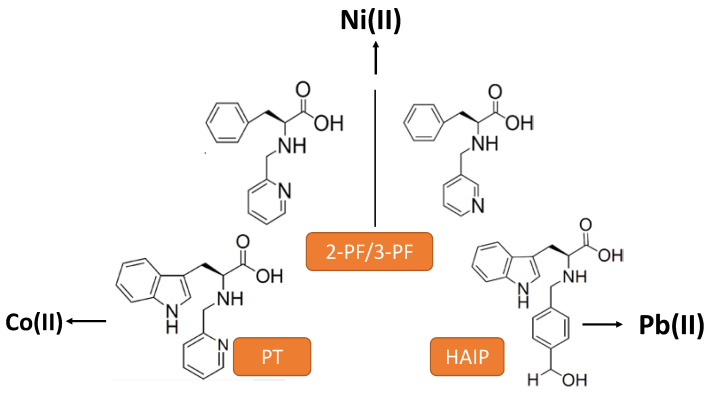
Scheme of the different tryptophan and phenylalanine derivatives used to synthesize new metal ion-induced hydrogels.

**Figure 20 nanomaterials-11-01865-f020:**
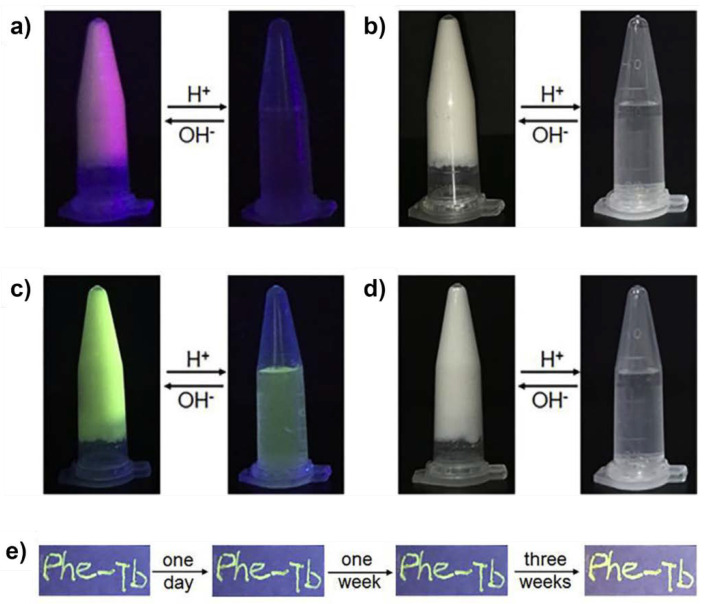
Images of luminescent ON/OFF behavior of Phe-Eu (**a**) and Phe-Tb (**c**) hydrogels subjected to pH stimuli under UV light; gel-sol transition images of Phe-Eu (**b**) and Phe-Tb (**d**) hydrogels exposed to pH stimuli under daylight; Photostability of the handwritten patterns “Phe-Tb” exposed to sunlight for different time (**e**). Adapted from reference [59] with permission, copyright 2018 Elsevier.

**Figure 21 nanomaterials-11-01865-f021:**
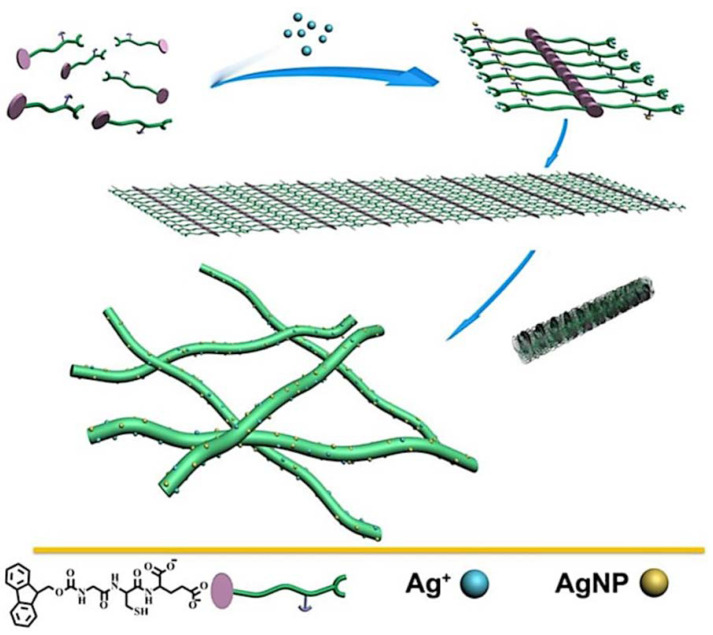
Proposed mechanism of the Fmoc-GCE self-assembly with Ag^+^. Reproduced from reference [61] with permission, copyright 2015 American Chemical Society.

**Figure 22 nanomaterials-11-01865-f022:**
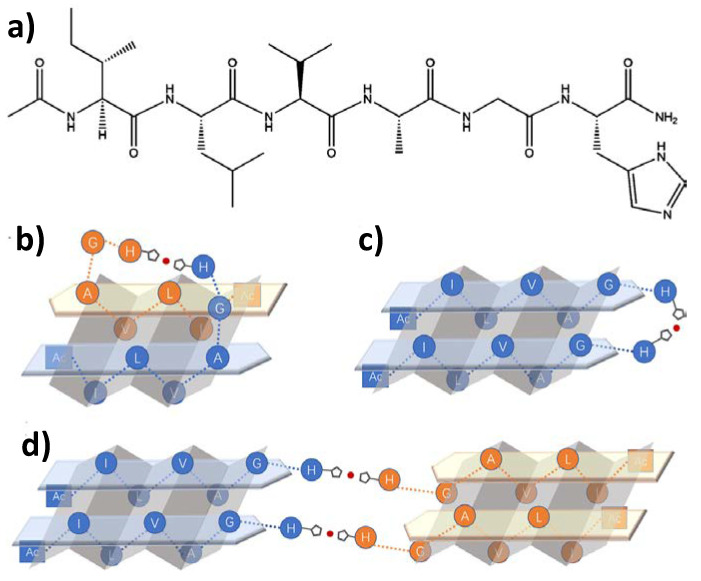
IH6 chemical structure (**a**). Scheme of the two possible assembly modes, antiparallel (**b**) and hairpin-like parallel (**c**), of two IH6 peptides coordinated to Ag^+^ through imidazole residues; possible intermolecular parallel mode (**d**). Adapted from reference [62] with permission, copyright 2018 American Chemical Society.

**Figure 23 nanomaterials-11-01865-f023:**
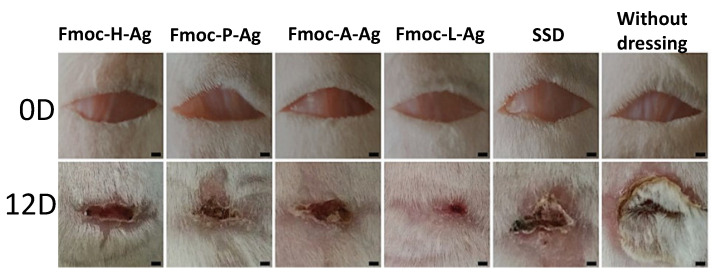
Photographs of the In Vivo assays of the different metallohydrogels as treatment for wound healing against the infection with *S. aureus* for 0 and 12 days (Where H = histidine, P = proline, A = alanine, L = leucine and SSD = silver sulfadiazine cream). Adapted from reference [63].

**Figure 24 nanomaterials-11-01865-f024:**
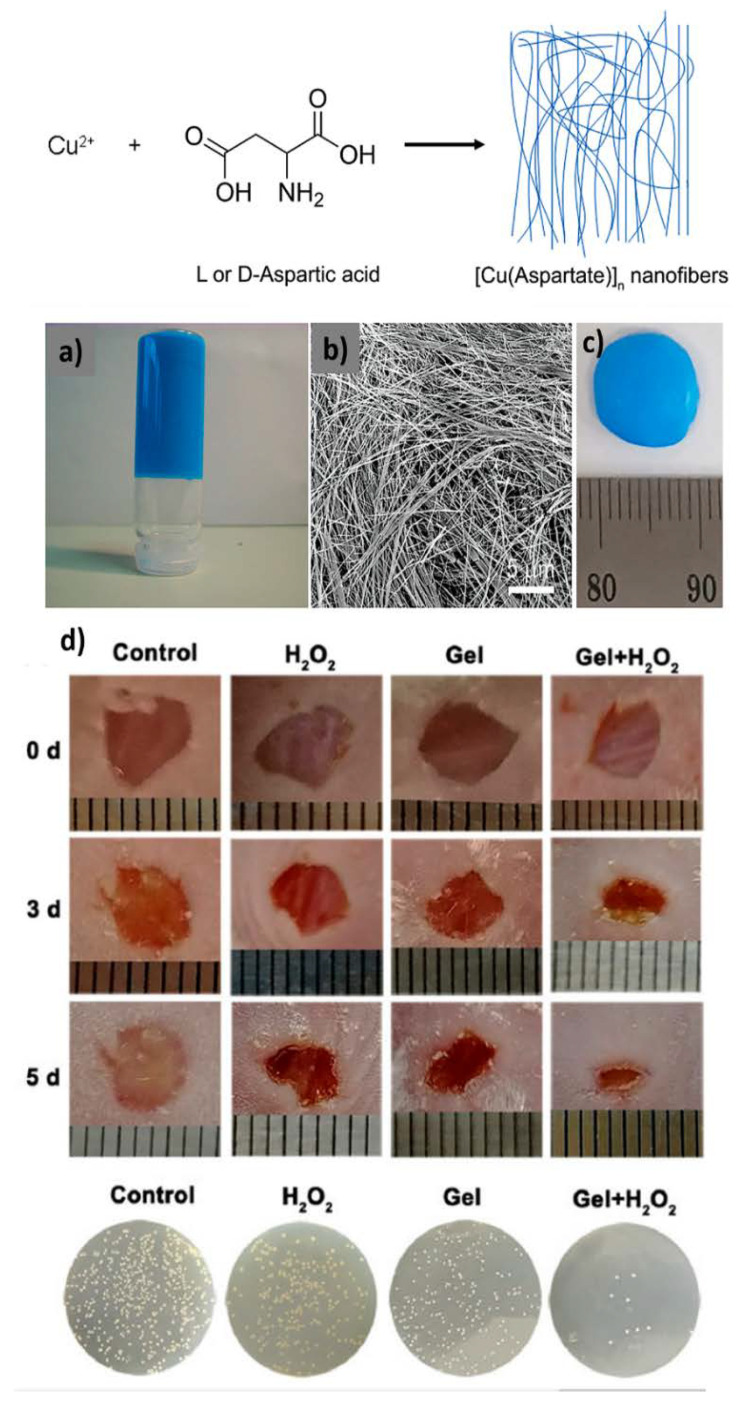
Illustration of the synthesis of Cu-Asp-based nanofibers; (**a**) Photograph and (**b**) FESEM image of the Cu-Asp-based gel and (**c**) photograph of hydrogel-based artificial enzyme. Digital images of infectious wounds and bacterial colonies formed by the bacteria in the wound tissues (**d**). Adapted from reference [25,66] with permission, copyright 2009 and 2020 American Chemical Society and Springer.

**Figure 25 nanomaterials-11-01865-f025:**
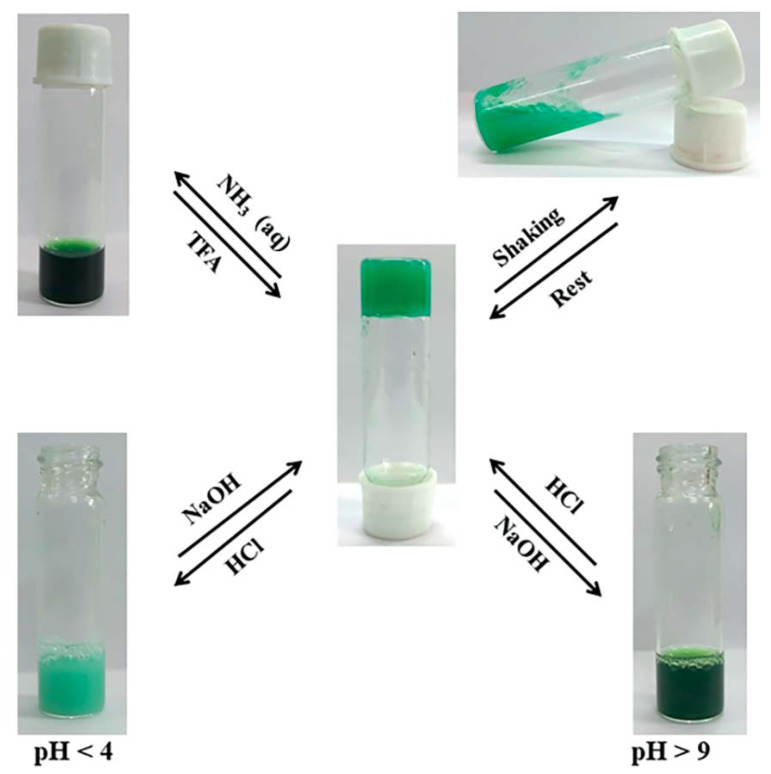
Schematic illustration of Cu-MOG in response to different stimuli. Adapted from reference [11] with permission, copyright 2019 John Wiley and Sons, Inc.

**Figure 26 nanomaterials-11-01865-f026:**
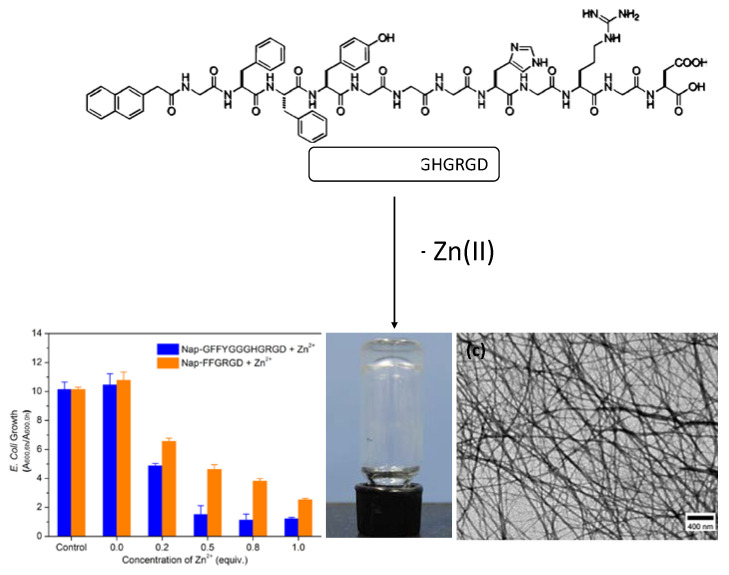
Schematic illustration of Zn-NapGFFYGGGHGRGD gel; their antibacterial growth inhibition (**a**), and its macroscopic (**b**) and microscopic morphology by scanning electron microscopy (SEM) (**c**). Adapted from reference [67] with permission, copyright 2015 The Author(s) and John Wiley and Sons, Inc.

**Figure 27 nanomaterials-11-01865-f027:**
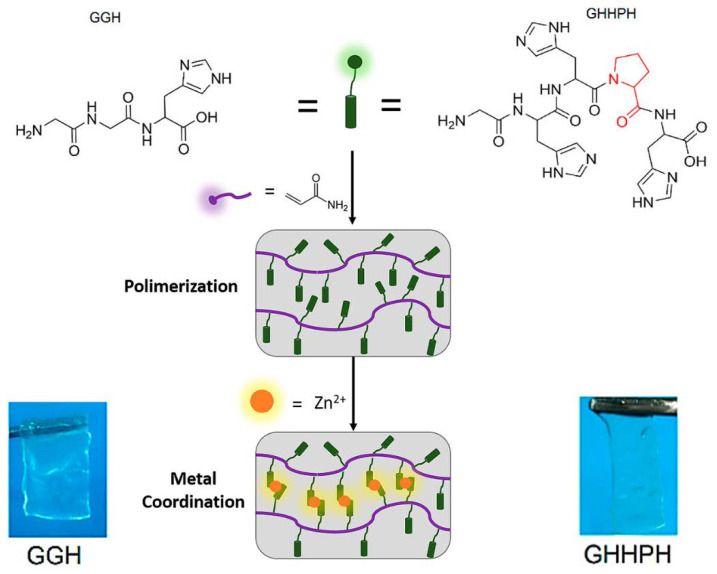
Scheme of polymerization and interaction process between GGH (left) and GHHPH (right) peptides, and Zn^2+^ ions to form the corresponding hydrogels. Adapted from reference [68] with permission, copyright 2019 The Author(s) and John Wiley and Sons, Inc.

**Figure 28 nanomaterials-11-01865-f028:**
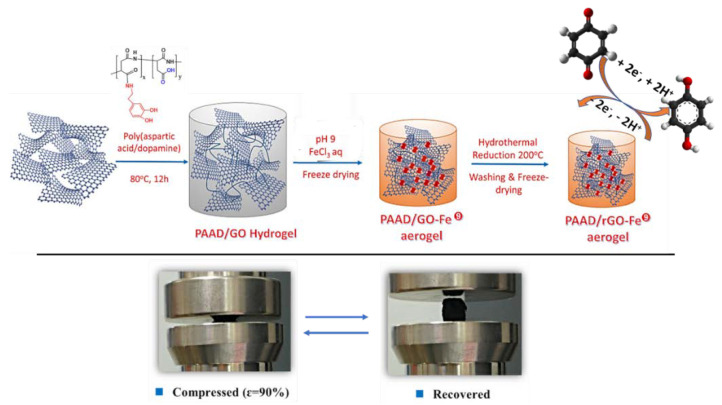
Scheme of the PAAD-rGO-Fe aerogel and its simplified mechanism of the reversible redox reaction (top); Photographs of the aerogel after compression at a strain of 90% (bottom). Adapted from reference [74] with permission. Copyright 2018 Springer Nature.

**Figure 29 nanomaterials-11-01865-f029:**
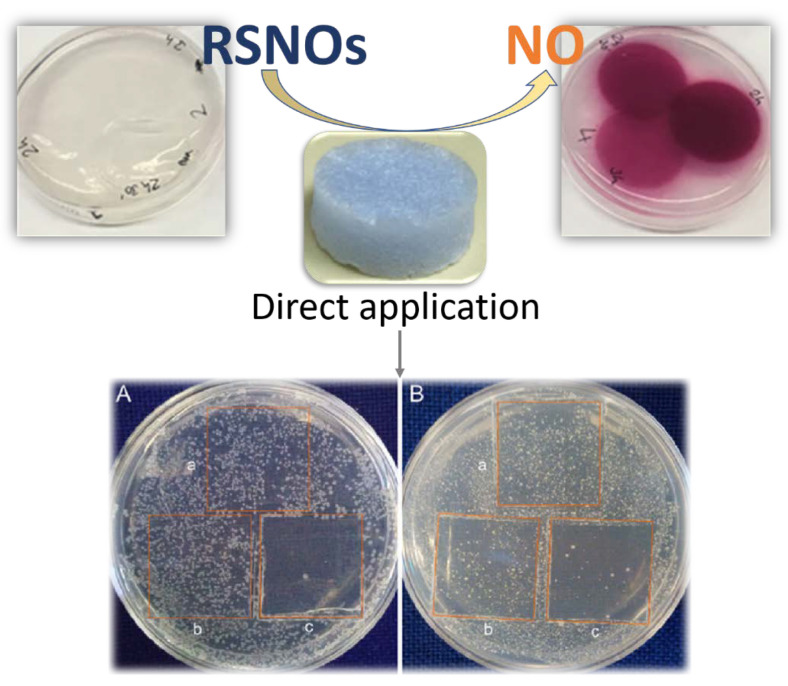
Schematic illustration Cu-cysteine/cellulose aerogel composite-where it releases NO from the decomposition of cysteine-NO (CysNO) by the composite material detected by the pink color appearing in agar plates modified with the Griess reagent and its antibacterial activity through growth inhibition of *E. coli* (**A**) and *S. epidermidis* (**B**) by releasing of NO. Number of colonies obtained after applying (a) CysNO, (b) CysNO + cellulose-PAH film, and (c) Cu-CysNO-cellulose + PAH film. Adapted from reference [75] with permission, copyright 2020 Elsevier.

**Figure 30 nanomaterials-11-01865-f030:**
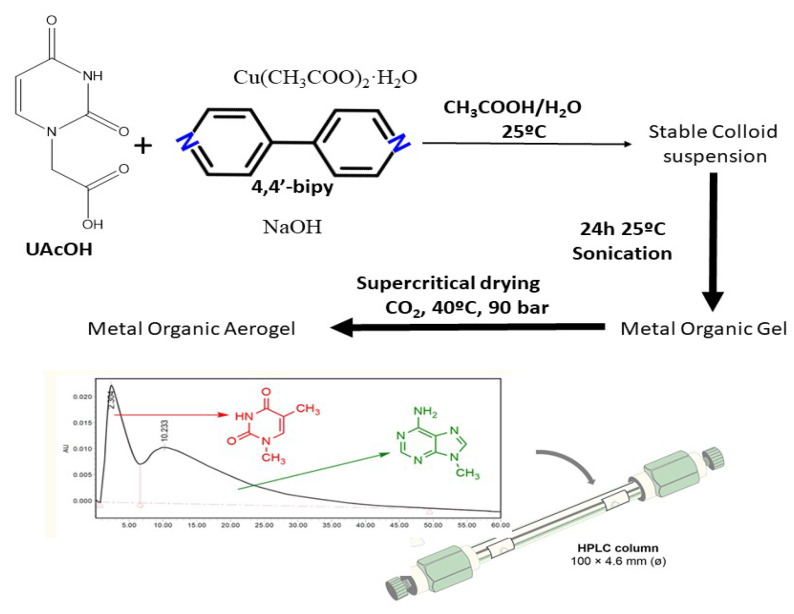
Schematic illustration of the reaction conditions to obtain the colloid, the Metal Organic hydrogel, and the Metal Organic Aerogel forms (**up**). Chromatogram obtained upon thymine and adenine separation using the quartz/metal–organic aerogel composite high-performance liquid chromatography (HPLC) column (**down**).

**Table 1 nanomaterials-11-01865-t001:** Rheological properties of the different metallohydrogels (coordination polymer gels (CPGs)-metal–organic gels (MOGs)) based on their storage and loss moduli values at constant oscillation strains.

CPGs/MOGs	Range of Frequency	Constant Oscillation Strain (%)	Approx. Loss Moduli, G″ (Pa)	Approx. Storage Moduli, G′ (Pa)	References
Zn(II)-AMP	0.03–30 Hz	0.4	10^4^	10^5^	[35]
Dy(III)-AMP	0.1–100 Hz	0.2	10^4^	10^5^	[25]
Ag(I)- inosine 5′monophsphate (IMP)	0.05–100 rad/s	1	~10	>55	[38]
Ag(I)-A	0.1–100 s^−1^	1	3 × 10^2^	10^3^	[37]
Cd(II)–T	0.05–100 rad/s	1	25	100	[39]
Cd(II)–U			15	80	
Zn(II)–cytosine	0.05–100 rad/s	1	3	20	[13]
Zn(II)–guanine			1.5	10	

**Table 2 nanomaterials-11-01865-t002:** Rheological properties of the different metallohydrogels (CPGs/MOGs) based on their storage and loss moduli values at constant oscillation strains.

CPGs/MOGs		Range of Frequency	Constant Oscillation Strain (%)	Approx. Loss Moduli, G″ (Pa)	Approx. Storage Moduli, G′ (Pa)	References
Ni(II)-derived tyrosine	P1	1–100 rad/s	0.1	35	200	[47]
	P2			10^3^	3 × 10^3^	
	P3			2 × 10^3^	0.5 × 10^4^	
Pt(II) nanoparticles-peptide(Phe) bolamphiphile		0.01–100 Hz	-	2 × 10^4^	1.7 × 10^5^	[48]
Pt(II) nanoparticles-peptide(Tyr) bolamphiphile				3 × 10^4^	1.5 × 10^5^	
Zn(II)-H_2_mugly		0.1–10 rad/s	0.5	13	23	[50]
Mg(II)-alanine derivative		0.1–100 rad/s	0.1	25	30	[52]
Cu(II)-amino succinic acid derivative (Cu-MOG)		1–100 rad/s	0.5	20	100	[11]
Co(II)-BTC-Arginine				10^4^	1	[57]
Ag(I)-Fmoc-GCE		0.1–10 Hz	-	<2 × 10^4^	>2 × 10^4^	[61]
Ag(I)-IH6		0.1–100 rad/s	0.1	-	4 × 10^4^	[62]
Ag(I)-Fmoc-amino acids		1–100 rad/s	1	50–100	200–1000	[63]
Zn(II)-GGH		0.01–100 rad/s	0.1	10^3^	10^3^	[68]
Zn(II)-GHHPH		0.01–100 rad/s	0.1	550	10^3^	

**Table 3 nanomaterials-11-01865-t003:** Porosity properties, density and applications of different metal–organic aerogels (MOAs).

MOAs	Specific Surface Area (m^2^ g^−1^)	Pore Size (μm)	Density (mg cm^−3^)	Simplicity of the Synthesis	Application	References
polyaspartic acid + dopamine, PAAD-graphene oxide, rGO-Fe	83.65	0.01–0.2	18.6	complex	Supercapacitors	[74]
Cu-cysteine/cellulose	-	4–130	41	complex	Nitric oxide delivery	[75]
Cu-modified Uracil	21	0.05–0.2	32.9	simple	Stationary phase in HPLC	[76]

**Table 4 nanomaterials-11-01865-t004:** Summary of coordination compounds-based colloids, gels and aerogels and its applications.

Metal Center	Biological Interest Ligands		Material State	Applications	References
	Nucleotides/nucleobases derivatives	Aminoacids/Peptides derivatives			
Ag(I)	GMP		Gel	Immobilization of protein	[24]
	A, C		Gel	Antibacterial	[37]
	IMP		Gel	Antibacterial, catalyst and water treatment	[38]
		Fmoc-GCE-OH	Gel	Sol-gel stimulus-response, catalyst and antibacterial	[61]
		IH6	Gel	Selective killing of wound-gressing/antibacterial	[62]
		Fmoc-Amino acids	Gel	Drug-delivery/antibacterial	[63]
Ln(III)	AMP		Gel	Sol-gel stimulus-response and encapsulation of glucose oxidase enzyme.	[25]
Cu(II)	TAcOH		Colloid	3D printing ink and humidity sensor	[17,18]
		Aspartic acid	Gel	Catalyst, antibacterial, wound-healing agent	[25,66]
	UAcOH		Aerogel	Stationary phase for HPLC column	[76]
		Phenylalanine based-amphiphiles	Gel	Encapsulation of dyes and vitamin B12 molecules	[49]
Zn(II)	AMP		Gel	Sol-gel stimulus-response and self-healing	[35]
		GFFYGGGHGRGD	Gel	Antimicrobial	[67]
Cd(II)	T, U		Gel	Template for the in-situ quantum dots growth	[39]
UiO-68	ATP		Gel	Controlled drug release	[43]
Ni(II)		Tyrosine-based amphiphiles	Gel	Sol-gel stimulus-response	[47]
Mn(II), Co(II), Ni(II)		Phenylalanine based-amphiphiles	Gel	Encapsulation of dyes and vitamin B12 molecules	[46]
Co(II)		BTC-arginine	Gel	Antibacterial/Sensor	[57]
Mg(II)		N-(7-hydroxyl-4-methyl-8-coumarinyl)-alanine	Gel	pH and mechano-responsive	[52]
Tb(III), Eu(III)		Phenylalanine	Gel	Luminescent inks for anti-counterfeiting	[59]
Fe(III)/Fe(II)		Polyaspartic acid	Aerogel	Supercapacitors	[74]
	AMP		Gel	Superparamgnetic/porosity	[36]

## Data Availability

Data is contained within the article.

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
