# Peer review of "Advances and Novel Perspectives on Colloids, Hydrogels, and Aerogels Based on Coordination Bonds with Biological Interest Ligands"

_nanomaterials, 2021, doi:10.3390/nano11071865_

Round 1

Reviewer 1 Report

In the manuscript, the authors overview the advances in colloids, gels, and aerogels containing  metal ions and ligands of biological interest. Moreover, they used some samples to briefly introduce their in applications in drug release, self-healable materials and sensors. Although authors summarized recent works, but it may be important to give more proposals and prospect.

Author Response

A1) Following the suggestion of reviewer 1, in the revised version, we have introduced several new proposals related to the objective of this perspective work. All changes have been highlighted in yellow. The examples include metal-organic gels with modified nucleotides and peptides with potential applications in water treatment, catalytic, and antimicrobial agents (references 36, 38, 57, 61, 62, 63, in the current version).

We have also added some complementary references that can broaden the vision of the work (actual reference numbers 58,64, 68)

Reviewer 2 Report

Dear Editor, in this review, the authors report some advances and novel perspectives of colloids, hydrogels, and aerogels based on coordination bonds with biological interest ligands. However, ithe references reported are not enaougt and recents to justify the interest of the presented manuscript.

Author Response

A1) We appreciate the comments of reviewer 2, but in this case, we disagree with his/her opinion in which he/she consider that the number of references used in the article is not sufficient nor recent enough to justify the interest of the work.

We want to emphasize that the article has 76 references, of which 67% are from the last five years. This data shows the current interest of the work and the increase in recent years of research related to colloids, gels, and aerogels with coordination bonds, transition metals, and bioinspired ligands.

In addition, 50% of the references used belong to international peer-reviewed journals of high impact factor, from 60.6 (Chem Rev. (4)), 54.4 (Chem. Soc. Rev. (5)); 21.6 (Nat. Chem. (1)); 20.8 (Coord Chem Rev. (2)); 18.8 Adv. Funct. Mat. (2)); 15.5 (J. Am. Chem. Soc.(2)); to 15.3 (Angew. Chem. (1)); Chem. Commun. (4); Chem. Mater. (3); Chem. A Eur. J (2); Nanoscale (2); Dalton. Trans. (4); ACS Appl. Nano Mater. (4), among others. These data show the interest of this area for a large number of researchers.

In any case, we have added eight more articles, recently published and closely related to the objective of the work to emphasize its interest. We understand that the exciting area of gels, colloids, and aerogels, will advance towards the use of bioinspired ligands due to the significant relevance that these ligands have from different points of view, both biological and biocompatible, biodegradable, low cost, etc...  Moreover, the synergy of these ligands with metal ions widens their properties and potential applications in little-explored fields such as the world of sensors, 3D printing or anticounterfeiting.

Reviewer 3 Report

Comments to the Author:

The present work “Advances and novel perspectives of colloids, hydrogels, and aerogels based on coordination bonds with biological interest ligands” demonstrate shows new advances in the synthesis of colloids, gels, and aerogels generated by combining metal ions and ligands of biological interest, such as nucleobases, nucleotides, peptides, or amino acids, among other derivatives. However, some interesting discussion and applications seem to be lost such as biobattery, biological fuel cell (BFC) or microbial fuel cell (MFC). Therefore, this work could not be published until some technical and fundamental issues are addressed.

  • Authors must organize Tables and rewrite discussion. The summarized Table for comparison is absent.
  • The author should add the energy application in the manuscript. It will clearly demonstrate the advantages of various colloids, hydrogels, and aerogels based on coordination bonds derivatives.

Author Response

Q1) Authors must organize Tables and rewrite discussion. The summarized Table for comparison is absent.

A1) Following the suggestion of reviewer 3, we have rewritten part of the discussion, expanded the tables, and added a summary table (table 4). All the modifications made are highlighted in yellow in the new version sent.

Q2) The author should add the energy application in the manuscript. It will clearly demonstrate the advantages of various colloids, hydrogels, and aerogels based on coordination bonds derivatives.

A2) Reviewer 3 rightly mentions that the topic of biofuels and biobatteries is of great interest. It should be noted that although it is true that colloids, gels, and aerogels based on coordination bonds are very interesting in the energy area, if we focus on these types of materials with bio-inspired ligands, the examples are drastically reduced.

Indeed, this topic is being investigated by using MOGs / MOAs but not with biological interest ligands (see references: Aerogels and metal–organic frameworks for environmental remediation and energy production Environmental Chemistry Letters (2018) 16:797–820 https://doi.org/10.1007/s10311-018-0723-x; Facile preparation of hierarchically porous carbons from metal-organic gels and their application in energy storage. SCIENTIFIC REPORTS | 3 : 1935 | DOI: 10.1038/srep01935). It is also being developed with ligands of biological interest without introducing coordination bonds to metal centers (see reference: Bioinspired nanoscale materials for biomedical and energy applications. J. R. Soc. Interface 11: 20131067. http://dx.doi.org/10.1098/rsif.2013.1067.). Therefore, the works published in this area are outside the focus of the work presented.

Following the suggestion of reviewer 3, and without losing sight of the objective of the work, we have added all the references published to date related to the use of colloids, MOGs / MOAs with biological interest ligands (nucleobases, amino acids, peptides, and derivatives)in the use of gas storage and catalysis, which is where there are some relevant examples.

In addition, in the conclusions, we have highlighted the interest of this approach that is still little explored. Changes made in the new version have been highlighted in yellow throughout the text.

Round 2

Reviewer 2 Report

Accept in present form

Reviewer 3 Report

The author has made adequate modification according to the comments, it can be accepted for publication in its current version.